# The vascular niche controls Drosophila hematopoiesis via fibroblast growth factor signaling

Manon Destalminil-Letourneau[†], Ismaël Morin-Poulard, Yushun Tian, Nathalie Vanzo, Michele Crozatier*

Centre de Biologie du Développement (CBD), Centre de Biologie Intégrative (CBI), Université de Toulouse, CNRS, UPS, Toulouse, France

**Abstract** In adult mammals, hematopoiesis, the production of blood cells from hematopoietic stem and progenitor cells (HSPCs), is tightly regulated by extrinsic signals from the microenvironment called 'niche'. Bone marrow HSPCs are heterogeneous and controlled by both endosteal and vascular niches. The Drosophila hematopoietic lymph gland is located along the cardiac tube which corresponds to the vascular system. In the lymph gland, the niche called Posterior Signaling Center controls only a subset of the heterogeneous hematopoietic progenitor population indicating that additional signals are necessary. Here we report that the vascular system acts as a second niche to control lymph gland homeostasis. The FGF ligand Branchless produced by vascular cells activates the FGF pathway in hematopoietic progenitors. By regulating intracellular calcium levels, FGF signaling maintains progenitor pools and prevents blood cell differentiation. This study reveals that two niches contribute to the control of Drosophila blood cell homeostasis through their differential regulation of progenitors.

*For correspondence:
michele.crozatier-borde@univ-tlse3.fr

Present address: [†]CellProthera, Paris, France

Competing interests: The authors declare that no competing interests exist.

## Introduction

In adult mammals, HSPCs in the bone marrow ensure the constant renewal of blood cells. The cellular microenvironment of HSPCs, called 'niche', regulates hematopoiesis under both homeostatic and immune stress conditions (*Asada et al., 2017*; *Calvi et al., 2003*; *Calvi and Link, 2015*; *He et al., 2014*; *Kiel et al., 2005*; *Kobayashi et al., 2016*; *Morrison and Scadden, 2014*; *Zhao and Baltimore, 2015*). Recent studies have revealed significant molecular and functional heterogeneity within the HSPC pool (for review *Haas et al., 2018*). These findings challenge the differential contribution of niche cell types to HSPC diversity. Given the high conservation of regulatory networks between insects and vertebrates, Drosophila has become an important model to study how hematopoiesis is controlled (*Evans et al., 2003*; *Hartenstein, 2006*). Insect blood cells, or hemocytes, are related to the mammalian myeloid lineage. In Drosophila, three blood cell types are produced: plasmatocytes that are macrophages involved in phagocytosis, crystal cells involved in melanisation and wound healing and lamellocytes required for encapsulation of pathogens too large to be destroyed by phagocytosis. Lamellocytes represent a cryptic cell fate since they only differentiate at the larval stage and in response to specific immune challenges such as wasp parasitism (*Lemaitre and Hoffmann, 2007*). The lymph gland is the larval hematopoietic organ and is composed of paired lobes, one pair of anterior lobes and several pairs of posterior lobes, aligned along the anterior part of the cardiac tube (CT) which corresponds to the vascular system (*Figure 1a* and *Lanot et al., 2001*). In third instar larvae, the anterior lobes comprise three zones: a medullary zone (MZ) containing hematopoietic progenitors, a cortical zone (CZ) composed of differentiated blood cells, and a small group of cells called the Posterior Signaling Center (PSC) (*Figure 1a* and *Crozatier et al., 2004*; *Jung et al., 2005*). The PSC produces a variety of signals that regulate lymph gland homeostasis (for

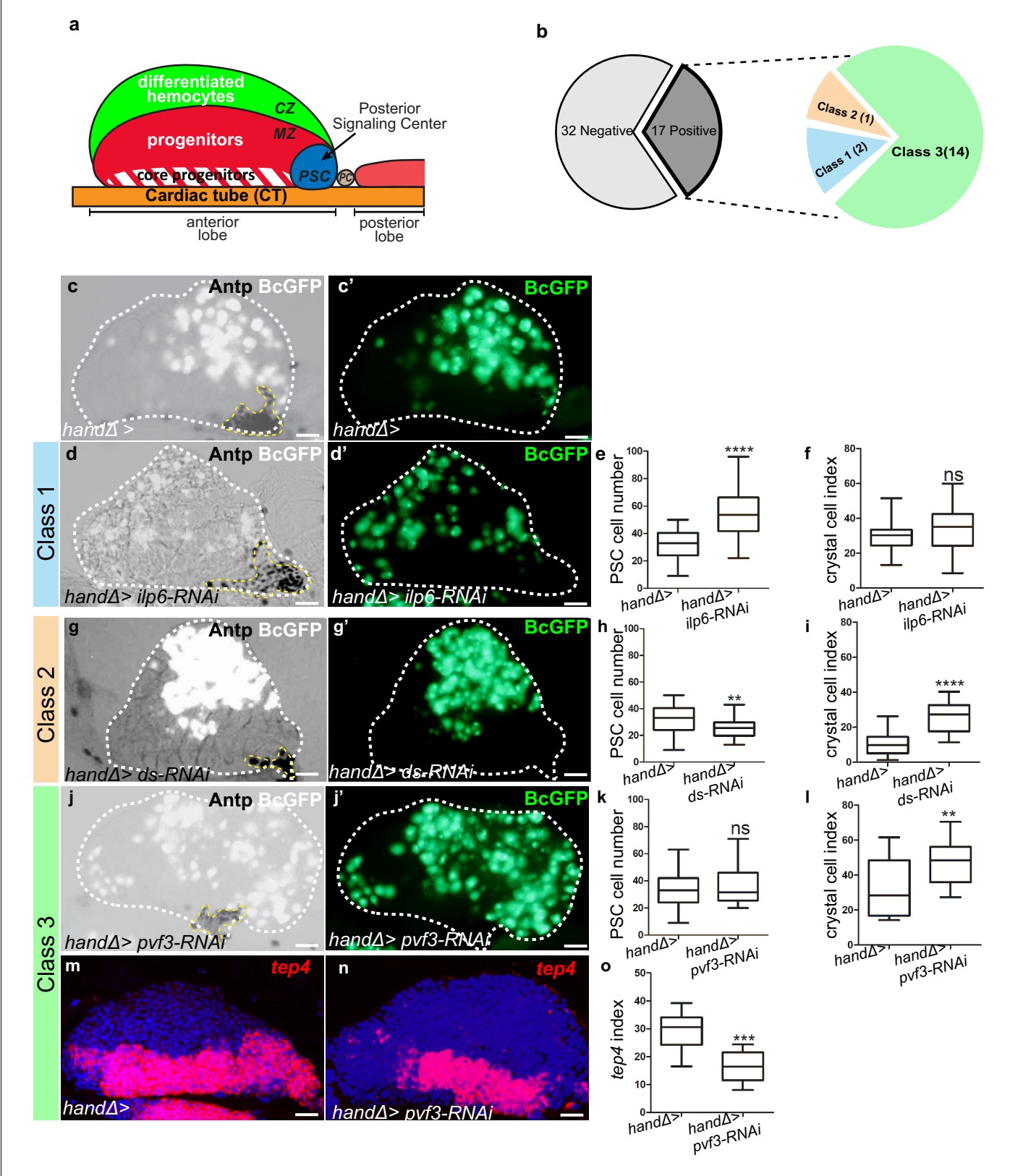

**Figure 1.** Lymph gland organization and RNAi screen results. (a) Representation of lymph gland anterior and posterior lobes from third instar larvae. The anterior lobe is composed of progenitors (red) and core progenitors (hatched red), and the cortical zone (CZ, green). The PSC is blue and the cardiac tube (CT)/vascular system, is orange. PC corresponds to pericardial cell. (b) Summary of the screen performed by expressing RNAi in

*Figure 1 continued on next page*

*Figure 1 continued*

cardiac cells using the *hand∆-gal4* driver. The number of genes corresponding to the different classes of phenotype is given. Subsequent panels illustrate the control and observed lymph gland defects (c, d, g, j). Anterior lobe and PSC are delimited by white and yellow dashed lines, respectively. Black-cell-GFP (BcGFP, white) labels crystal cells and Antp (black) the PSC. (c', d', g', j') BcGFP is in green; (e, h, k) PSC cell numbers; (f, i, l) Crystal cell index. (c–f) Reducing *ilp6* in cardiac cells (d, d') augments PSC cell number (e) without affecting crystal cell differentiation (f); this defines class 1. (g–i) Knocking down *dachsous (ds)* in cardiac cells (g, g') decreases PSC cell number (h) and increases crystal cell index (i); this defines class 2. (j–l) Reducing *pvf3* in cardiac cells (j, j') does not modify PSC cell number (k) but increases crystal cell differentiation (l); this defines class 3. (m, n) *tep4* (red) labels core progenitors. Decrease in *tep4* expression is observed when *pvf3* is knocked down in cardiac cells. (o) *tep4* index. For all quantifications and figures, statistical analysis *t*-test (Mann-Whitney nonparametric test) was performed using GraphPad Prism five software. Error bars represent SEM and *p<0,1; **p<0,01; ***p<0,001; ****p<0,0001 and ns (not significant). In all confocal pictures nuclei are labeled with Topro (blue) and scale bars = 20 μm.

The online version of this article includes the following source data and figure supplement(s) for figure 1:

**Source data 1.** Results of the RNAi ligand screen RNAi was expressed in cardiac cells by using the *hand∆-gal4* and/or *NP1029-gal4* driver.

**Source data 2.** Results of the RNAi ligand screen.

**Figure supplement 1.** Expression pattern of *hand∆-gal4* and *NP1029-gal4* driver in lymph glands during larval development.

**Figure supplement 1—source data 1.** RNAi screen quantification data.

review see *Banerjee et al., 2019*; *Letourneau et al., 2016*; *Yu et al., 2018*). Recently we established that cardiac cells produce the ligand Slit which, through the activation of Robo receptors in the PSC, controls the proliferation and clustering of PSC cells and in turn their function (*Morin-Poulard et al., 2016*). Furthermore, the MZ progenitor population is heterogeneous and a subset of progenitors called 'core progenitors' which express the *knot/Collier (Kn/Col)* and the *thioester-containing protein-4* (*tep4*) genes is aligned along the cardiac tube and are maintained independently from the PSC (*Figure 1a* and *Baldeosingh et al., 2018*; *Benmimoun et al., 2015*; *Oyallon et al., 2016*). Altogether, these data led us to ask whether signals derived from cardiac cells are involved in the control of lymph gland homeostasis, i.e. the balance between progenitors and differentiated blood cells, independently from the PSC. To address this possibility we performed a candidate RNAi screen in cardiac cells to identify new potential signaling pathways involved in the crosstalk between the vascular and the hematopoietic organs.

Here we show that several signals produced by cardiac cells contribute to maintain lymph gland homeostasis. We investigated in more detail the role of the Fibroblast Growth Factor (FGF) ligand Branchless (Bnl). FGF signaling is conserved during evolution and is less complex in Drosophila than in humans. Ligand binding to a FGF receptor (FGFR) promotes its dimerization, which results in its tyrosine-phosphorylation, thus providing a scaffold to recruit different partners (*Muha and Müller, 2013*; *Ornitz and Itoh, 2015*). In mammals, ligand binding to the FGFR activates Ras/Raf-Mek-MAPK, PI3K/AKT, and PLCγ-Ca$^{2+}$ signaling pathways (*Turner and Grose, 2010*).The Drosophila genome encodes two FGF receptors, Breathless (Btl) and Heartless (Htl), and three ligands, Bnl, Thisbe (Ths) and Pyramus (Pyr), (*Beiman et al., 1996*; *Glazer and Shilo, 1991*; *Gryzik and Müller, 2004*; *Klämbt et al., 1992*; *Sutherland et al., 1996*). Htl is activated by Ths and Pyr, while Btl is activated by Bnl. We established that Bnl is expressed in cardiac cells and signals to its receptor Breathless (Btl) expressed in progenitors. Bnl/Btl-FGF activation controls progenitor intracellular Ca$^{2+}$ concentration, probably by activating Phospholipase Cγ (PLCγ) which regulates endoplasmic reticulum Ca$^{2+}$ stores. Altogether, these data strongly support the conclusion that the cardiac tube plays a role similar to a niche by regulating lymph gland hematopoiesis.

## Results

### A cardiac screen identifies genes controlling lymph gland homeostasis

To investigate the role of cardiac cells in the control of lymph gland hematopoiesis, we performed a functional screen based on the expression, in cardiac cells, of RNAis directed against transcripts encoding known Drosophila ligands. For this we used the cardiac *hand∆-gal4* driver which is expressed in cardiac cells throughout the three larval stages (*Figure 1—figure supplement 1a–c'* and *Monier et al., 2005*; *Morin-Poulard et al., 2016*) to screen a collection of RNAi lines corresponding to 49 Drosophila ligands (*Figure 1—source data 2*). As read-outs, we analyzed blood cell differentiation with the crystal cell reporter BcGFP (*Tokusumi et al., 2009*), and PSC cell numbers

and morphology by performing Antennapedia (Antp) immunostaining (*Mandal et al., 2007*). Compared to the control, 17 RNAi lines showed lymph gland homeostasis defects that were classified into three groups (*Figure 1b*). Class 1: Increased PSC cell numbers but no effect on crystal cell differentiation (*Figure 1c–f*). Two RNAis against *ilp6* and *spätzle4* transcripts belong to this class (source data). Class 2: Decreased PSC cell numbers and increased crystal cell differentiation (*Figure 1g–i*). Only one RNAi against *dachsous* (*ds*) belongs to this class (source data) Class 3: No effect on PSC cell numbers but increased crystal cell differentiation (*Figure 1j–l*); 14 RNAis belong to this class. The class three phenotype strongly suggested that signals from cardiac cells could control crystal cell differentiation independently from the PSC. We extended the analysis of the 14 corresponding genes by labeling the core progenitors with *tep4* in situ hybridization (*Krzemień et al., 2007*). Reduced *tep4* expression was observed for 12 RNAi treatments out of 14 (*Figure 1m–o* andsource data), indicating that the corresponding genes are required in cardiac cells to maintain *tep4* expression in lymph gland progenitors and to prevent crystal cell differentiation. To avoid any bias due to the *handΔ-gal4* driver, we also tested *NP1029-gal4*, an independent cardiac cell driver (*Figure 1—figure supplement 1d–f'* and *Monier et al., 2005*; *Morin-Poulard et al., 2016*). Among the 14 RNAi candidates, nine gave a similar phenotype with both drivers (source data). In conclusion, our functional screen allowed us to identify nine ligands involved in communication between cardiac cells and hematopoietic progenitors to control lymph gland homeostasis.

## The FGF ligand Bnl from cardiac cells controls lymph gland homeostasis

One candidate identified in our screen was Bnl. Previous studies have shown that Htl-FGF signaling is required during both early embryogenesis for lymph gland specification (*Grigorian et al., 2013*; *Mandal et al., 2007*) and in L3 larvae to control lymph gland progenitors (*Dragojlovic-Munther and Martinez-Agosto, 2013*). However, no role for *bnl* in the lymph gland has been described yet. Since *bnl* knock-down in cardiac cells significantly enhanced crystal cell differentiation in the lymph gland we decided to pursue an analysis of the Bnl-FGF pathway. Since Bnl is a diffusible ligand, we first documented *bnl* mRNA expression by in situ hybridization. *bnl* is expressed in cardiac and pericardial cells (*Figure 2a–a''*), in agreement with previously published data (*Jarecki et al., 1999*). We also observed a weak *bnl* expression in MZ progenitors (as labeled by domeMESO >GFP in *Figure 2a–a''*), in differentiating hemocytes (as labeled by hml >GFP) and in a subset of crystal cells (marked by BcGFP) whereas no expression was detected in the PSC (*Figure 2—figure supplement 1a–c''*). In a heterozygous *bnl* loss-of-function mutant context where one copy of *bnl* (*bnl^{P2}/+*) is missing, we observed an increased number of crystal cells compared to the control (*Figure 2b–d*). To specifically knock down *bnl* in cardiac cells, we expressed *bnl-RNAi* under the control of the cardiac tube specific driver *handΔgal4*. *bnl* loss-of-function experiments were performed from the L2 larval stage on, after the cardiac tube had formed, to avoid possible cardiac tube morphological defects (see MM and *Figure 2—figure supplement 1d–e*). *bnl* down-regulation in cardiac cells resulted in increased differentiation of both crystal cells and plasmatocytes (*Figure 2e–f,i* and *Figure 2—figure supplement 1f–h*). Increased crystal cell differentiation was also observed using another independent *bnl*-RNAi line (*Figure 2—figure supplement 1i–k*) and with the alternative *NP1029-gal4* driver (*Figure 2—figure supplement 1l–n*). Applying *bnl* knockdown only after the L2 stage by using the GAL80 ^{ts} system (*McGuire et al., 2004*) led to a similar crystal cell differentiation defect (*Figure 2—figure supplement 1o–q*). We then analyzed MZ progenitors when *bnl* was knocked down in cardiac cells, using DomeMESO-RFP that labels all progenitors, and *tep4* and Col that are expressed in the core progenitors (*Krzemień et al., 2007*; *Oyallon et al., 2016*). Compared to wild type, a reduced expression of the three markers was observed in *handΔ>bnl-RNAi* lymph glands (*Figure 2j–r*), indicating that Bnl from cardiac cells non-cell autonomously controls MZ progenitor maintenance. Altogether, these data indicate that Bnl produced in the cardiac tube acts in third instar larvae to control lymph gland homeostasis.

Since *bnl* is transcribed in MZ progenitors, though at low levels, we also analyzed its function in these cells. Reduction of *bnl* expression in progenitors (*dome >bnl-RNAi)* led to a significant increase in crystal cell differentiation as well as a decrease in *tep4* expression (*Figure 2—figure supplement 2a–f*), indicating that Bnl produced by MZ progenitors is required to maintain their identity and to prevent their differentiation. Altogether, these data show that Bnl is produced by both MZ and cardiac cells, and that both sources are required in the control of lymph gland homeostasis.

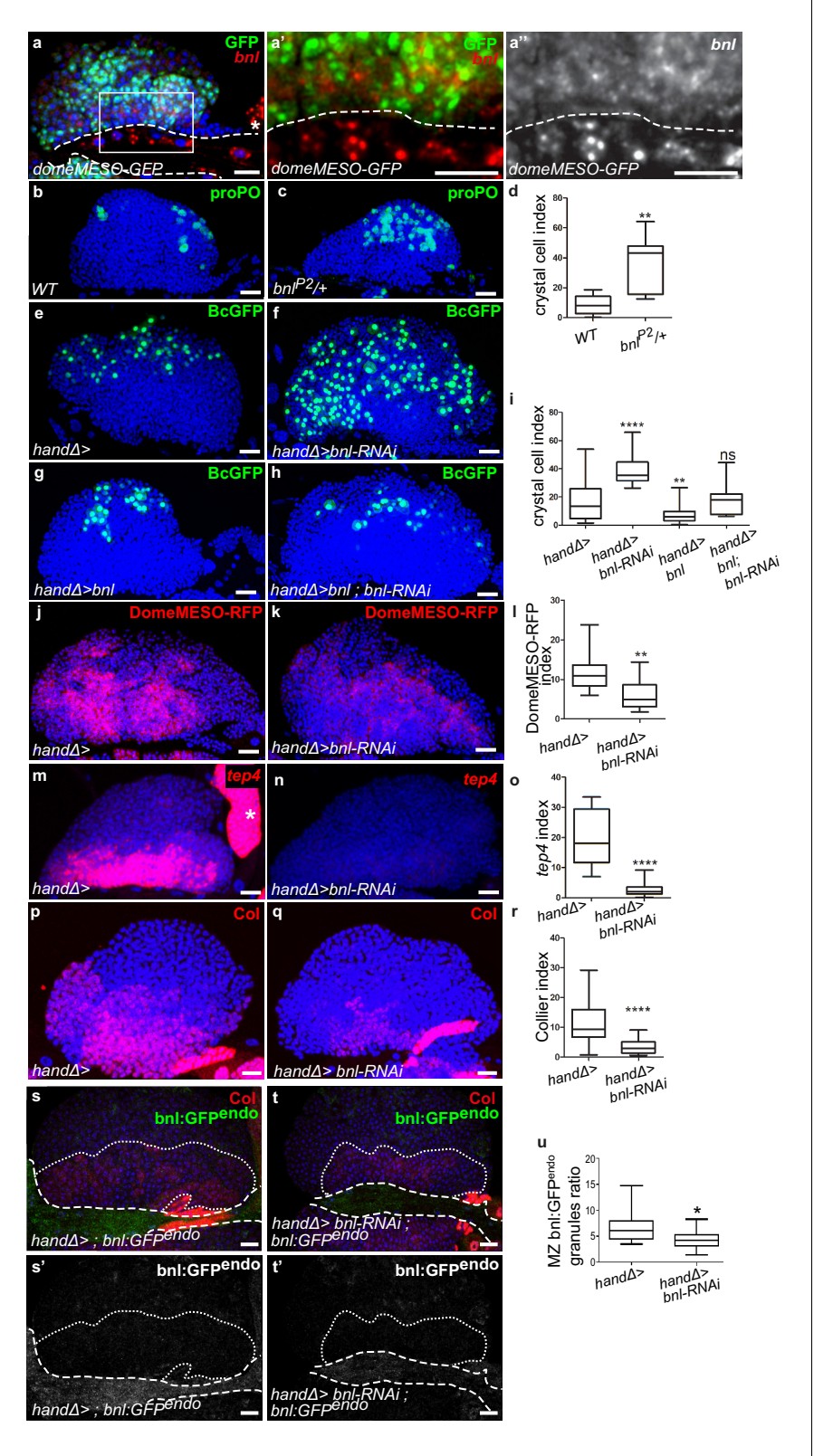

**Figure 2.** Ligand Bnl is expressed in cardiac cells and controls lymph gland homeostasis. (a) A maximum projection of 5 confocal lymph gland sections, *bnl* (red) is expressed in cardiac cells and MZ progenitors that express domeMESO-GFP (green). (a', a'') An enlarged view, *bnl* is red (a') or white (a''). A white dashed line indicates the cardiac tube. * indicates a pericardiac cell. (b, c) proPO (green) labels crystal cells. *bnl^{P2}/+* heterozygous mutant lymph glands have an increased number of crystal cells (c) compared to the control (b). (e–f, g–h) Black-cell GFP (BcGFP, green) labels crystal cells. (d, i)

*Figure 2 continued on next page*

*Figure 2 continued*

Crystal cell index. Co-expression of *bnl* and *bnl-RNAi* in cardiac cells restores the wildtype number of crystal cells (i). (j, k) DomeMESO-RFP (red) labels MZ progenitors. Compared to the control (j) barely detectable DomeMESO-RFP levels are observed when *bnl* is knocked down in cardiac cells (k). (l) DomeMESO-RFP index. (m, n) *tep4* labels core progenitors. Compared to the control (m) lower levels of *tep4* (red) are observed when *bnl* is knocked down in cardiac cells (n). (o) *tep4* index. (p–q) Col labels core progenitors. Compared to the control (p) lower levels of Col are observed in the core progenitors when *bnl* is knocked down in cardiac cells (q). (r) Col index. (s–t') Maximum projection of 5 confocal sections of the lymph gland expressing *bnl:GFP* $^{endo}$ (green) and Col immunostaining that labels MZ progenitors (red). Compared to the control (s, s') a decrease in bnl:GFP $^{endo}$ in green (t) and white (t') is observed when *bnl* is knocked down in cardiac cells. Fine and thick dashed lines indicate the MZ and CT contours, respectively. (u) Bnl:GFP$^{endo}$ granules ratio in the MZ.

The online version of this article includes the following source data and figure supplement(s) for figure 2:

**Source data 1.** Quantification of *Figure 2*.
**Figure supplement 1.** The ligand Bnl in cardiac cells controls lymph gland hemocyte differentiation homeostasis.
**Figure supplement 1—source data 1.** Quantification of *Figure 2—figure supplement 1*.
**Figure supplement 2.** Whereas the ligand Bnl in cardiac cells does not control PSC cells, it is required in MZ progenitors to regulate lymph gland homeostasis.
**Figure supplement 2—source data 1.** Quantification of *Figure 2—figure supplement 2*.

To determine whether Bnl produced by cardiac cells contributes to the pool of Bnl present in the MZ, we analyzed endogenous Bnl distribution. We used the bnl:GFP$^{endo}$ knock-in allele that recapitulates *bnl* expression (*Du et al., 2018*). In agreement with in situ bnl detection, bnl:GFP$^{endo}$ was found in cardiac cells and in MZ progenitors (*Figure 2s–s'*). However, when *bnl* was knocked down only in the cardiac tube (*hand∆>bnl-RNAi, bnl:GFP$^{endo}$*; *Figure 2t–u*) we overserved a reduction of *bnl:GFP$^{endo}$* both in cardiac cells and MZ progenitors thus establishing that Bnl produced by cardiac cells contributes to the global MZ Bnl pool. The concomitant increased hemocyte differentiation suggests that Bnl levels contributed by the heart are required for lymph gland homoeostasis. To further support this conclusion, we overexpressed *bnl* only in cardiac cells (*hand∆>bnl)* which led to reduced crystal cell numbers (*Figure 2i*), and this confirms that the level of Bnl produced by cardiac cells controls hemocyte differentiation. Finally, rescue of crystal cell numbers (*Figure 2g–i*) and of progenitor marker expression (*Figure 2—figure supplement 2g–i*) was observed with a simultaneous expression of *bnl* and *bnl-RNAi* in cardiac cells (*hand∆>bnl; bnl-RNAi*). Altogether, these data establish that Bnl produced by cardiac cells is required for lymph gland hematopoiesis.

Since the PSC controls lymph gland cell differentiation (*Benmimoun et al., 2015*; *Morin-Poulard et al., 2016*; *Oyallon et al., 2016*; *Tokusumi et al., 2010*), we also looked at PSC cells by analyzing the expression of the PSC marker Antp (*Mandal et al., 2007*) when *bnl* was downregulated in cardiac cells. No PSC cell number or clustering defects were observed (*Figure 2—figure supplement 2j–l*). Hh expression in the PSC regulates progenitors and blood cell differentiation (*Baldeosingh et al., 2018*; *Mandal et al., 2007*; *Tokusumi et al., 2010*). The *hhF4-GFP* reporter transgene (*Tokusumi et al., 2010*) was expressed in PSC cells in the *hand∆>bnl*-RNAi context similar to the control (*Figure 2—figure supplement 2m–o*). These data strongly suggest that cardiac cell Bnl neither affects PSC cell numbers nor Hh activity, and likely acts directly on MZ progenitors to control lymph gland homeostasis. Altogether, these data indicate that although it is transcribed in many lymph gland cells, *bnl* expression in cardiac cells plays an essential role in the control of lymph gland homeostasis.

## The FGF receptor Btl expressed in progenitors, controls lymph gland homeostasis

Bnl activates the FGF pathway by binding Btl (*Kadam et al., 2009*). To document endogenous Btl expression in the lymph gland, we used the btl:cherry$^{endo}$ knock-in allele which recapitulates Btl expression (*Du et al., 2018*). Strong btl:cherry$^{endo}$ expression was observed in cardiac cells and lower levels in MZ progenitors (as labeled by domeMESO-GFP in *Figure 3a–a''*). Whereas no expression was detected in PSC cells, a very faint expression occurs in a subset of crystal cells and in most differentiating blood cells (labeled by BcGFP and Hml >GFP, respectively, in *Figure 3—figure supplement 1a–c''*).

To determine whether *btl* is required for lymph gland homeostasis, we looked at crystal cell differentiation in a heterozygous loss-of-function mutant context, where one copy of *btl* is mutated

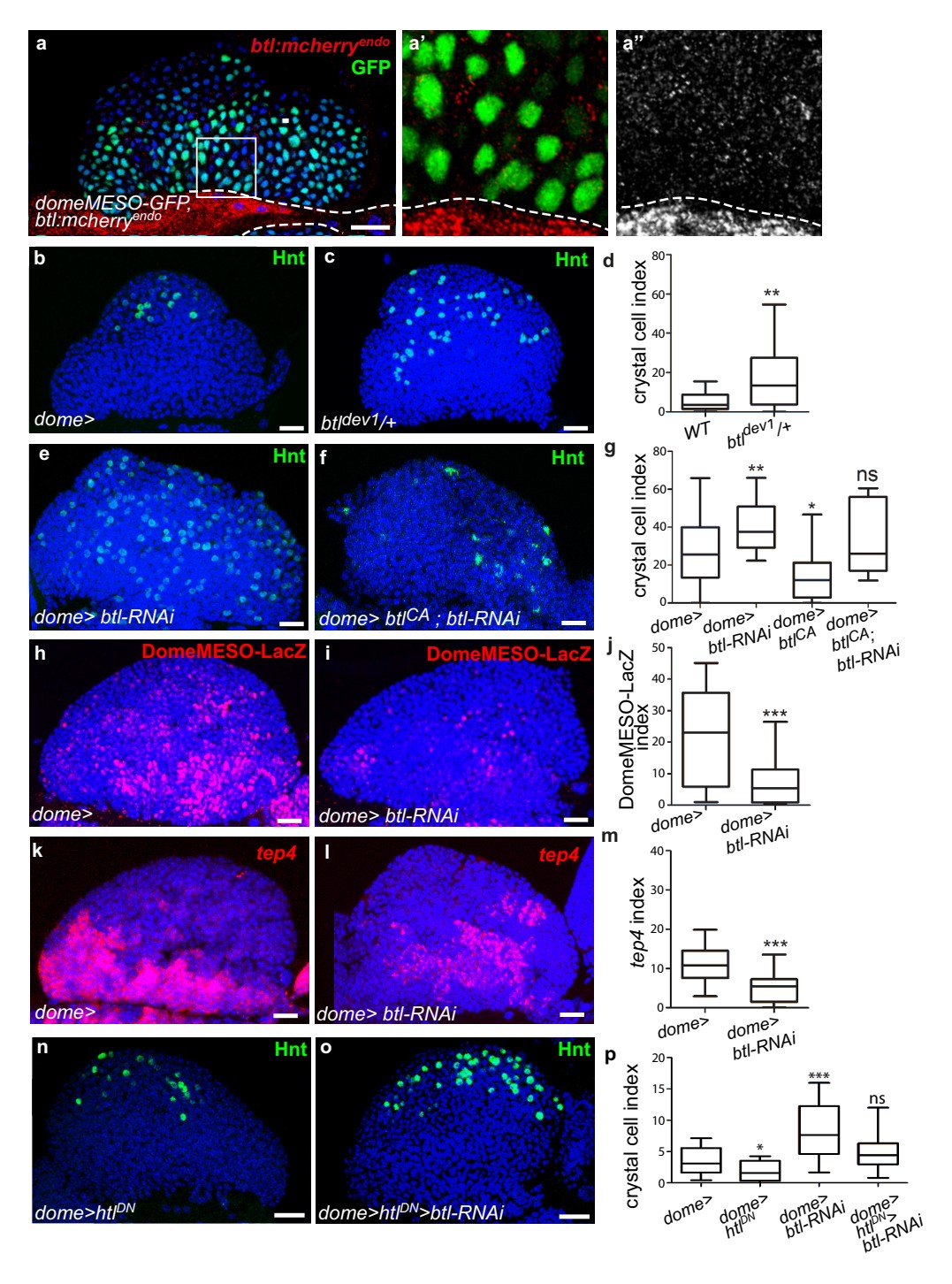

**Figure 3.** Receptor Btl is expressed in hematopoietic progenitors and required to control lymph gland homeostasis. (a) A maximum projection of 5 confocal lymph gland sections of larvae expressing *btl:cherry^endo* (red) and *domeMESO-GFP* that labels MZ progenitors (green). (a', a'') An enlarged view, *btl:cherry^endo* red (a') or white (a''). Dashed lines indicate the cardiac tube contour. *btl:cherry^endo* is expressed in cardiac cells and MZ progenitors. (b–c, e–f) Hindsight (Hnt, green) labels crystal cells. Crystal cell differentiation is increased in *btl^dev1/+* heterozygous mutant larvae (c) compared to the control (b). (e, f) Crystal cell numbers increase when *btl* is knocked down in progenitors (e) and crystal cell differentiation is rescued when a constitutive activated *btl* receptor (*btl^CA*) is expressed in the *btl-RNAi* context (f). (d, g) Crystal cell index. (h, i) DomeMESO-LacZ (red) labels MZ progenitors. Compared to the control (h) barely detectable domeMESO-LacZ levels are observed when *btl* is knocked down in progenitors (i). (k, l) Lower levels of *tep4* (red) are observed when *btl* is knocked down in progenitors (l) compared to the control (k). (j, m) DomeMESO-LacZ and tep4 index, respectively.
*Figure 3 continued on next page*

eLife Research article

Developmental Biology

(*btl^dev1^/+*). The resulting crystal cell index was higher than in controls (*Figure 3b–d*), revealing that *btl* controls lymph gland homeostasis. We then knocked down *btl* by expressing RNAi in either MZ progenitors (*dome>*) or cardiac cells (*handΔ>*). *btl* downregulation in cardiac cells did not significantly affect crystal cell numbers or MZ progenitors (*Figure 3—figure supplement 1d–i*). In contrast, knocking down *btl* in MZ progenitors led to increased crystal cell (*Figure 3e,g*) and plasmatocyte numbers (*Figure 3—figure supplement 1j–l*), together with a reduced expression of the two progenitor markers *domeMESO-LacZ* and *tep4* (*Figure 3h–m*). We then performed rescue experiments with a constitutively active form of Btl (*btl^CA^*, *Parés and Ricardo, 2016*). The expression of *btl^CA^* in progenitors (*dome >btl^CA^*) led to reduced crystal cell numbers, i.e., a phenotype opposite to that of *btl* loss-of-function (*Figure 3g*). The co-expression of *btl^CA^* and *btl-RNAi* in progenitors (dome >*btl*-RNAi>*btl^CA^*; *Figure 3f,g*) rescued crystal cell differentiation, confirming that Btl is required in MZ progenitors. Finally, we examined whether PSC cells were affected when *btl* was knocked down in progenitors. No difference in PSC cell numbers or clustering was observed compared to the control (*Figure 3—figure supplement 1m–o*). We conclude that *btl* expression in MZ progenitors is required to control lymph gland homeostasis.

As opposed to Btl-FGF inhibition, Htl-FGF pathway knock-down in MZ progenitors was reported to block blood cell differentiation (*Dragojlovic-Munther and Martinez-Agosto, 2013*). To investigate the relationship between the two pathways in the MZ, we performed epistasis experiments. Expression of a dominant–negative form of Htl (*dome >Htl^DN^*) in progenitors led to a decrease in crystal cell differentiation, in agreement with a previous report (*Dragojlovic-Munther and Martinez-Agosto, 2013*). Simultaneous expression of Htl^DN^ and *btl-RNAi* (*dome >Htl^DN^ >* btl-RNAi) restored a wildtype number of crystal cells (*Figure 3n–p*). These data indicate that there is no hierarchy between Btl-FGF and Htl-FGF pathways. They also suggest that their simultaneous activity in MZ progenitors ensures a robust regulation of hemocyte differentiation.

In conclusion, downregulating Btl in MZ progenitors causes a defect in lymph gland homeostasis similar to that caused by Bnl downregulation in cardiac cells. This strongly suggests that Bnl/Btl-FGF signaling mediates inter-organ communication between MZ progenitors and the vascular system. This leads us to propose that by acting directly on MZ progenitors, the cardiac tube plays a role similar to a niche.

## Bnl secreted by cardiac cells is taken up by lymph gland progenitors

Bnl originating from cardiac cells and acting on MZ progenitors raised the question of its mode of diffusion. To investigate this question, we expressed a functional GFP-tagged version of Bnl (*UAS-Bnl::GFP, Lin, 2009*) in cardiac cells. In addition to the expected GFP detection in these cells, discrete GFP positive cytoplasmic punctate dots/granules were detected in MZ progenitors (*Figure 4a–a''*), indicating that Bnl::GFP can propagate from cardiac to lymph gland cells. Many cytoplasmic Bnl-GFP positive punctate dots in the MZ were Btl:Cherry positive (*Figure 4b–b''*). To further characterize these Bnl-GFP dots, we labeled recycling vesicles and late endosomes, using the ubi-Rab11-cherryFP reporter and Rab7 immunostaining, respectively. We found that many Rab11-positive and Rab7-positive vesicles co-localized with Bnl-GFP in the MZ (*Figure 4c–d''*). The simplest explanation is that Bnl::GFP secreted by cardiac cells is internalized by MZ progenitors, likely through receptor-mediated endocytosis.

To further confirm the role of Bnl secreted by cardiac cells, we impaired endoplasmic reticulum (ER) vesicle formation by knocking down the Secretion-associated Ras-related GTPase1 (Sar1) specifically in cardiac cells. The Sar1GTPase plays a key role in the biogenesis of transport vesicles and acts by regulating vesicular trafficking (*Saito et al., 2017*; *Yorimitsu et al., 2014*). The simultaneous

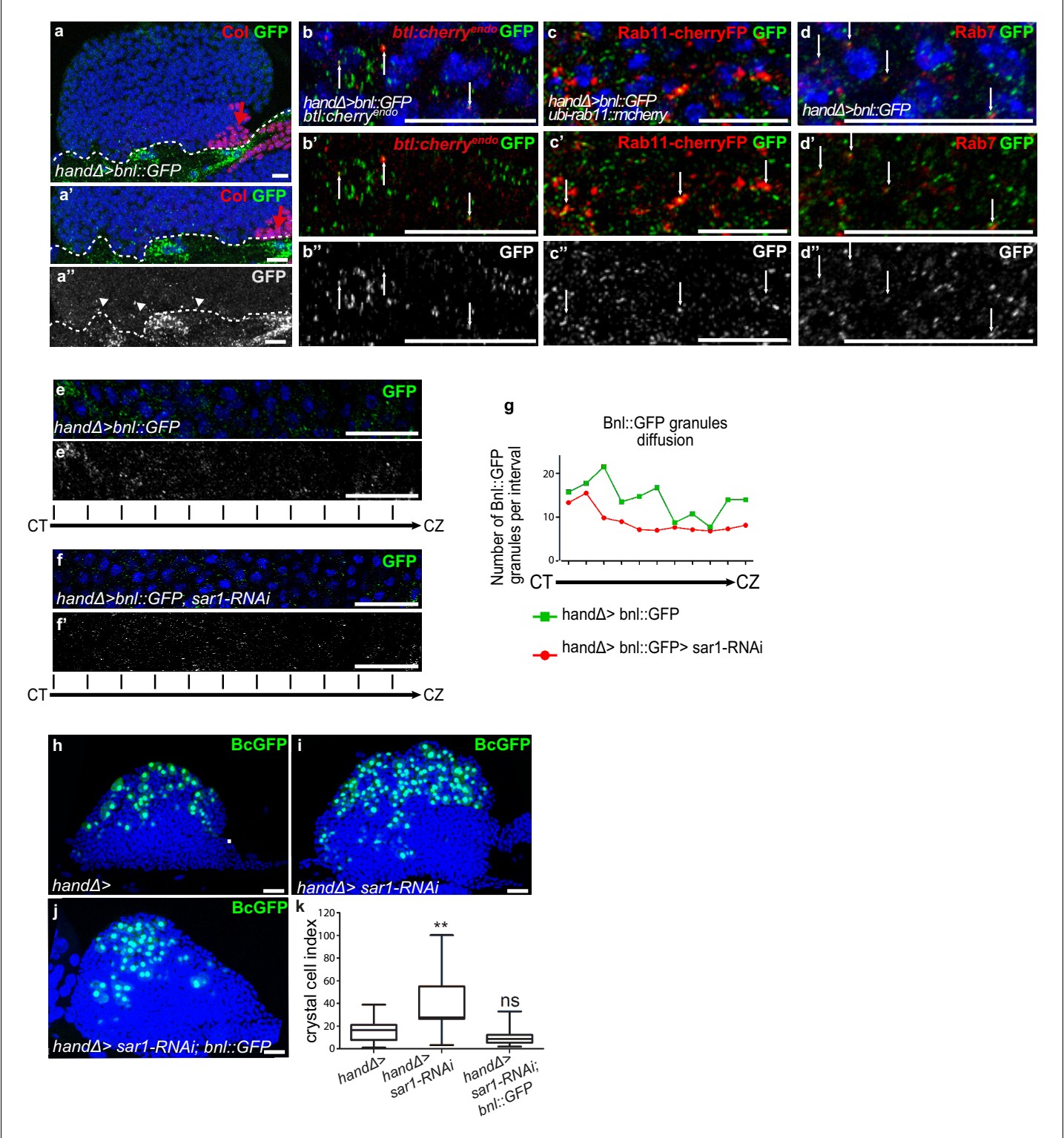

**Figure 4.** Ligand Bnl secreted by cardiac cells controls lymph gland crystal cell differentiation. (a) Active bnl::GFP fusion protein is expressed in cardiac cells using *hand∆-gal4* driver. Dashed lines indicate cardiac tube and the PSC is labeled by Collier (Col, red and red arrow). (a', a'') An enlarged view; Bnl::GFP is green (a') or white (a''). Bnl::GFP positive granules are detected in cardiac and lymph gland cells (arrowheads). (b–b") Enlargement of MZ area close to the cardiac tube in larvae expressing Bnl::GFP fusion protein (green) in cardiac cells (*hand∆-gal4 > Bnl::GFP*) and Btl:mcherry[endo] (red). Bnl::GFP cytoplasmic punctate dots (green in b-b' and white in b') co-localize with Btl:mcherry[endo] (yellow and arrows in b'). (c,c") Enlargement of MZ area close to the cardiac tube in larvae expressing *ubi-Rab11cherryFP* (red), a marker for recycling endocytic vesicles; Bnl::GFP fusion protein (green) is

*Figure 4 continued on next page*

*Figure 4 continued*

expressed in cardiac cells (*hand∆-gal4 > Bnl::GFP*). Bnl::GFP cytoplasmic punctate dots (green in c-c' and white in c') co-localize with ubi-Rab11cherryFP (yellow and arrows in c'). (**d–d"**) Enlargement of MZ area close to the cardiac tube in larvae expressing Bnl::GFP fusion protein (green) in cardiac cells (*hand∆-gal4 > Bnl::GFP*) and Rab7 immunostainings (red in d, d' and white in d'). (**d–d'**) Bnl::GFP cytoplasmic punctuate dots co-localize with Rab7 positive dots (yellow and arrows in d'). (**e–f'**) Enlargement of lymph gland cross sections extending from the cardiac tube (CT) to the cortical zone (CZ). Bnl::GFP fusion protein, expressed in cardiac cells (*hand∆-gal4 > Bnl::GFP*) is green (e, f) and white (e', f'). Knocking down *sar1* in cardiac cells (f, f') leads to a decrease in Bnl::GFP cytoplasmic punctate dots compared to the control (e, e'). (**g**) Quantification of Bnl::GFP cytoplasmic punctate dots/ granules. (**h, j**) BcGFP (green) labels crystal cells. Knocking down *sar1* in cardiac cells (i) increases crystal cell numbers compared to the control (h). Crystal cell differentiation rescue is observed when bnl::GFP is co-expressed with *sar1-RNAi* (j, k). (**k**) Crystal cell index.

The online version of this article includes the following source data and figure supplement(s) for figure 4:

**Source data 1.** Quantification of *Figure 4*.
**Figure supplement 1.** Knocking down *sar1* in cardiac cells impairs crystal cell differentiation and increases PSC size.
**Figure supplement 1—source data 1.** Quantification of *Figure 4—figure supplement 1*.

expression of *sar1-RNAi* and Bnl::GFP in cardiac cells using the *hand∆-gal4* driver (*hand∆>bnl:: GFP >sar1* RNAi) resulted in reduced bnl::GFP cytoplasmic punctate dots in MZ progenitors compared to the control (*Figure 4e–g*), further confirming that cardiac cells secrete Bnl. Previous data had established that the Slit ligand, produced by cardiac cells, activates Robo receptors in the PSC and as a consequence controls PSC cell proliferation and clustering (*Morin-Poulard et al., 2016*). Consistent with the impairment of cardiac cell secretion when *sar1* is knocked down in cardiac cells (*hand∆>and NP1029>sar1* RNAi, *Figure 4—figure supplement 1a–f*), we also observed an increase in PSC cell numbers as well as a slight defect in their clustering, likely an effect of Slit/Robo impairment. These data indicate that *sar1* knock-down in cardiac cells impairs their secretory capacity and therefore Bnl secretion.

We then analyzed the consequences on lymph gland blood cell differentiation. When *sar1* was knocked down in cardiac cells using *hand∆* or NP1029 drivers (*Figure 4h–i,k* and *Figure 4—figure supplement 1g–i*) crystal cell numbers were higher than the control, indicating that cardiac cell secretion capacity is necessary for lymph gland hematopoiesis. To determine whether the overexpression of *bnl* in cardiac cells can compensate for decreased secretion, we performed rescue experiments. Crystal cell differentiation was improved when *sar1-RNAi* and bnl::GFP were simultaneously expressed in cardiac cells (*Figure 4j,k*). In conclusion, these results indicate that Bnl secreted by cardiac cells is likely taken up by MZ progenitors to activate Btl-FGF signaling, which in turn regulates lymph gland homeostasis.

## FGF activation in progenitors regulates their calcium levels

The next step was to address how Bnl/Btl-FGF signaling in progenitors controls lymph gland homeostasis. Depending on the cellular context, the FGF pathway activates the MAPK or PI3K pathways, or PLCγ that controls intracellular $Ca^{2+}$ levels (*Ornitz and Itoh, 2015*). Inactivation of MAPK or PI3K in lymph gland progenitors leads to a phenotype opposite to that of knock-down of *btl* in progenitors or of *bnl* in cardiac cells (*Dragojlovic-Munther and Martinez-Agosto, 2013*). This strongly suggests that Bnl/Btl-FGF signaling in progenitors does not involve MAPK or PI3K activity. It was previously reported that intracellular $Ca^{2+}$ levels regulate hematopoietic progenitor maintenance: reduction of cytosolic $Ca^{2+}$ in lymph gland progenitors leads to the loss of progenitor markers and to increased blood cell differentiation (*Shim et al., 2013*). Since Bnl/Btl-FGF knock-down and reduction of $Ca^{2+}$ in progenitors induce similar lymph gland defects, we asked whether both mechanisms were functionally linked. We investigated $Ca^{2+}$ levels within MZ progenitors using the $Ca^{2+}$ sensor GCaMP3, which emits green fluorescence only with high $Ca^{2+}$ levels (*Nakai et al., 2001*). This sensor is expressed under the control of the *dome-gal4* driver. In agreement with previous reports, high $Ca^{2+}$ levels were detected in MZ progenitors (*Figure 5a* and *Shim et al., 2013*). Knocking down *btl* in MZ progenitors starting from L1 stage led in third instar larvae to decreased fluorescence compared to the control, indicating a reduction in $Ca^{2+}$ levels (*Figure 5a–c*). Since no difference with control larvae could be observed at L2 stage (*Figure 5—figure supplement 1a–c*), we conclude that the Bnl/ Btl-FGF pathway is not required for MZ progenitor specification but is required in third instar larvae to regulate $Ca^{2+}$ levels. We then asked whether restoring high $Ca^{2+}$ levels in progenitors could rescue the lymph gland defects due to reduced Bnl/Btl-FGF function. When free $Ca^{2+}$ was high within

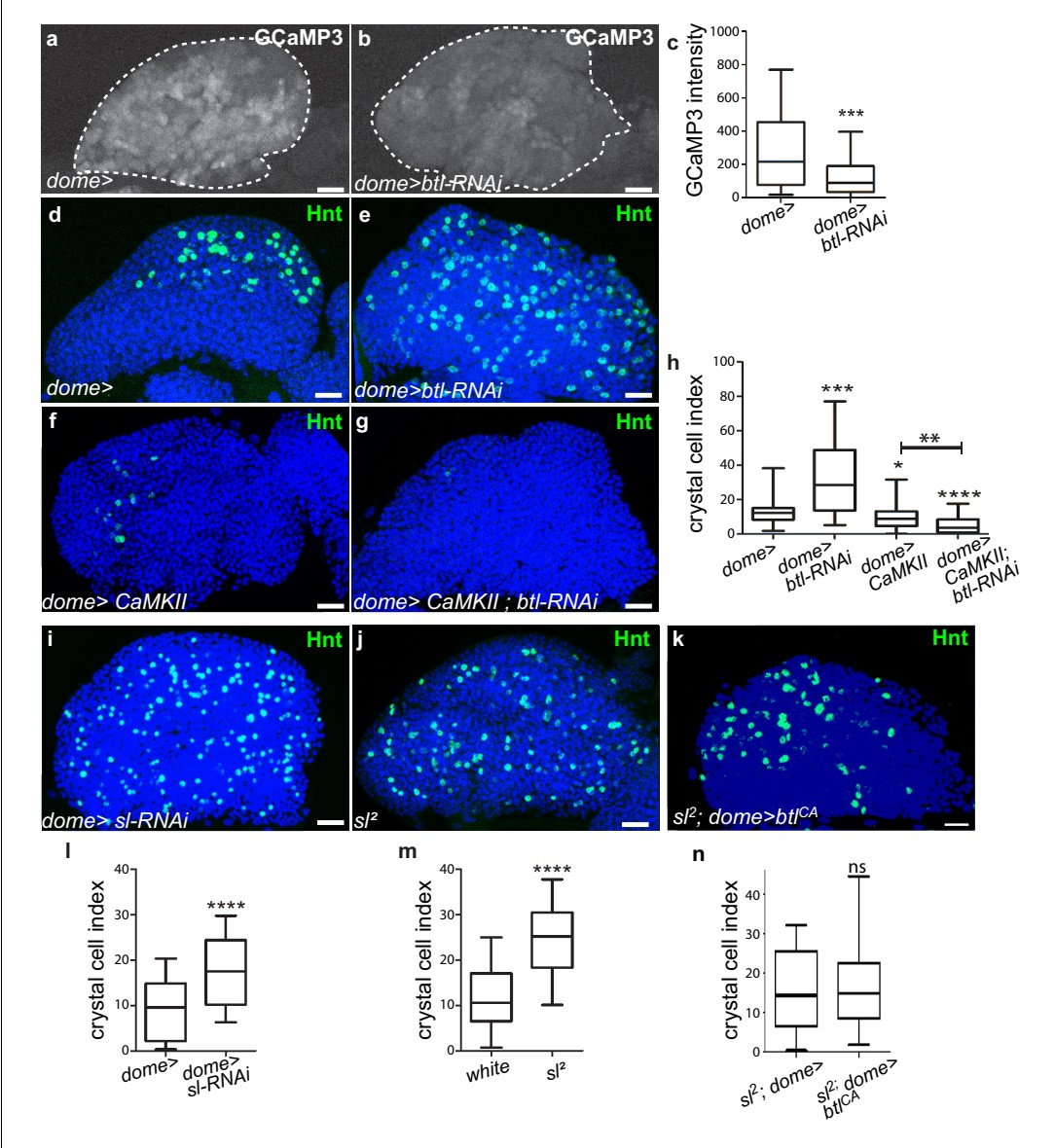

**Figure 5.** Btl receptor interacts genetically with CamKII to control blood cell differentiation by preventing high $Ca^{2+}$ levels in progenitors. (a, b) GCaMP3 $Ca^{2+}$ sensor (*dome >UAS-GCaMP3*) is white. GCaMP3 intensity decreases when *btl is* knocked down in MZ progenitors (**b**) compared to the control (**a**). (**c**) Quantification of GCaMP3 intensity. (**d–g, i–k**) Hnt (green) labels crystal cells. Crystal cell differentiation decrease is observed when $Ca^{2+}$ levels increased due to CaMKII expression in progenitors (*dome >CaMKII*, **f**) compared to the control (**d**). Co-expression of CaMKII and *btl*-RNAi in progenitors (*dome >CaMKII; btl-RNAi*, **g**) leads to a decrease in crystal cell number compared to the *btl* knock-down alone (**e**). (**h, l–n**) Crystal cell index. Crystal cell differentiation increase is observed in $sl^2$ homozygous mutant larvae (**j, m**) and when *sl* is knocked down in progenitors (**i, l**) compared to the control (**d**). (**k, n**) No difference in crystal cell index is observed in $sl^2$ homozygous mutant larvae and in a $sl^2$ homozygous mutant where $btl^{CA}$ is expressed in MZ progenitors ($sl^2$; *dome >btl^{CA}$).

The online version of this article includes the following source data and figure supplement(s) for figure 5:

**Source data 1.** Quantification of *Figure 5*.

**Figure supplement 1.** $Ca^{2+}$ levels in progenitors regulate crystal cell differentiation.

**Figure supplement 1—source data 1.** Quantification of *Figure 5—figure supplement 1*.

the progenitors, such as following overexpression of Calmodulin dependent kinase II (CaMKII; *dome >CaMKII*) or of IP3R (Tep4 >UAS-IP3R) which controls ER-mediated $Ca^{2+}$ release into the cytosol (*Shim et al., 2013*), a significant decrease in crystal cell numbers was observed compared to the control (*Figure 5d–h* and *Figure 5—figure supplement 1d–h*). This indicates that high $Ca^{2+}$ levels in

progenitors inhibit blood cell differentiation, which is in agreement with previously published data (*Shim et al., 2013*). Simultaneous *btl* reduction and Ca$^{2+}$ increase in progenitors, through CaMKII or IP3R overexpression (*dome >CamKII;btl-RNAi Figure 5g–h* and *tep4 >IP3R>btl-RNAi; Figure 5— figure supplement 1g–h*), leads to a decrease in crystal cell numbers compared to the sole *btl-RNAi* knockdown. Overall, these data suggest that in MZ progenitors, regulation of Ca$^{2+}$ levels functions downstream of the Bnl/Btl-FGF pathway. In vertebrates, FGF activation can recruit and activate PLCγ, which induces Ca$^{2+}$ release from the ER (*Ornitz and Itoh, 2015*). *small wing* (*sl*) encodes the sole Drosophila PLCγ homolog (*Thackeray et al., 1998*). To investigate the role of PLCγ in the lymph gland, we analyzed crystal cell differentiation in strong hypomorphic *sl²* mutants. Compared to the control, increased crystal cell differentiation was observed (as labeled by Hnt *Figure 5j,m*). We then addressed the role of *PLCγ* specifically in progenitors. Knocking down PLCγ in progenitors (*dome >sl-RNAi*) led to increased crystal cell differentiation compared to the control (*Figure 5i,l*), revealing that PLCγ in MZ progenitors regulates lymph gland hemocyte differentiation. Finally, we performed epistasis experiments to decipher the relationship between *sl* and the Btl-FGF pathway. When *btl$^{CA}$* was expressed in progenitors in a *sl²* mutant context (*sl²; dome >blt$^{CA}$; Figure 5k,n*), the crystal cell index was similar to that in the *sl²* mutant alone. These data establish that *sl* acts downstream of the Bnl/Btl-FGF pathway. Altogether, our data support the hypothesis that in MZ progenitors, Bnl/Btl-FGF signaling leads to the activation of PLCγ, which controls Ca$^{2+}$ levels and in turn hemocyte differentiation.

## Discussion

The control of HSPCs by a specific microenvironment called 'niche' is established both in mammals and in Drosophila. The niche is defined by its capacity to directly regulate, through signals, stem cells and progenitors. In the mammalian bone marrow HSPCs are under the control of the endosteal and vascular niches (*Asada et al., 2017*; *Calvi et al., 2003*; *Calvi and Link, 2015*; *He et al., 2014*; *Kiel et al., 2005*; *Morrison and Scadden, 2014*). In Drosophila, lymph gland studies have so far concentrated on the PSC acting as a niche. However, a subset of lymph gland progenitors (core progenitors), which express Col and *tep4* and are aligned along the cardiac tube, is maintained in the lymph gland even when the PSC function is impaired, suggesting that other signals alongside those from the PSC are required (*Baldeosingh et al., 2018*; *Benmimoun et al., 2015*; *Oyallon et al., 2016*). Here, we report that cardiac cells play a role similar to a niche, since they directly control core progenitor maintenance. We show that communication between the vascular system and the lymph gland involves Bnl/Btl-FGFsignaling. Bnl secreted by cardiac cells activates Bnl/Btl-FGF in progenitors, which in turn controls hemocyte homeostasis. Our data indicate that Bnl/ Btl-FGF signaling regulates lymph gland homeostasis by controlling calcium levels in progenitors via PLCγ activation (*Figure 6*). In a previous study, we showed that signals from the cardiac tube, namely Slit, can act on the PSC, but that no cellular communication between the cardiac tube and MZ progenitors is involved (*Morin-Poulard et al., 2016*). Now we establish that cardiac cells regulate the extent of progenitor differentiation in the lymph gland. Therefore, two separate niches (the PSC and the cardiac tube) control lymph gland homeostasis. While the PSC acts only on a subset of MZ progenitors (*Baldeosingh et al., 2018*; *Oyallon et al., 2016*), the cardiac tube directly regulates core progenitors and in turn all MZ progenitors (*Figure 6*).The identification of two niches that differentially regulate lymph gland progenitors sheds further light on the parallels existing between Drosophila lymph gland and mammalian bone marrow hematopoiesis.

Btl-FGF signaling regulates trachea morphogenesis, which builds the insect respiratory system (*Glazer and Shilo, 1991*; *Klämbt et al., 1992*; *Muha and Müller, 2013*; *Sato and Kornberg, 2002*; *Sutherland et al., 1996*). How the ligand Bnl diffuses from its source to activate Btl in neighboring cells remains a controversial issue. Studies performed on Drosophila larval Air-Sac-Primordium (ASP), using endogenous tagged versions of Bnl and Btl, brought to light a key role of long range direct cellular contacts mediated by long thin cellular extensions called cytonemes (*Roy et al., 2011*; *Sato and Kornberg, 2002*). In this process, rather than diffusing passively, Bnl produced by wing disc cells is delivered directly to ASP cells by cytonemes to activate FGF signaling (*Du et al., 2018*). No cytoplasmic extensions from either cardiac cells or MZ progenitors were observed so far, ruling out the delivery of Bnl from cardiac cells though long cytoplasmic extensions. Instead, both cardiac cells and MZ progenitors are embedded in a dense network of extra-cellular matrix (ECM)

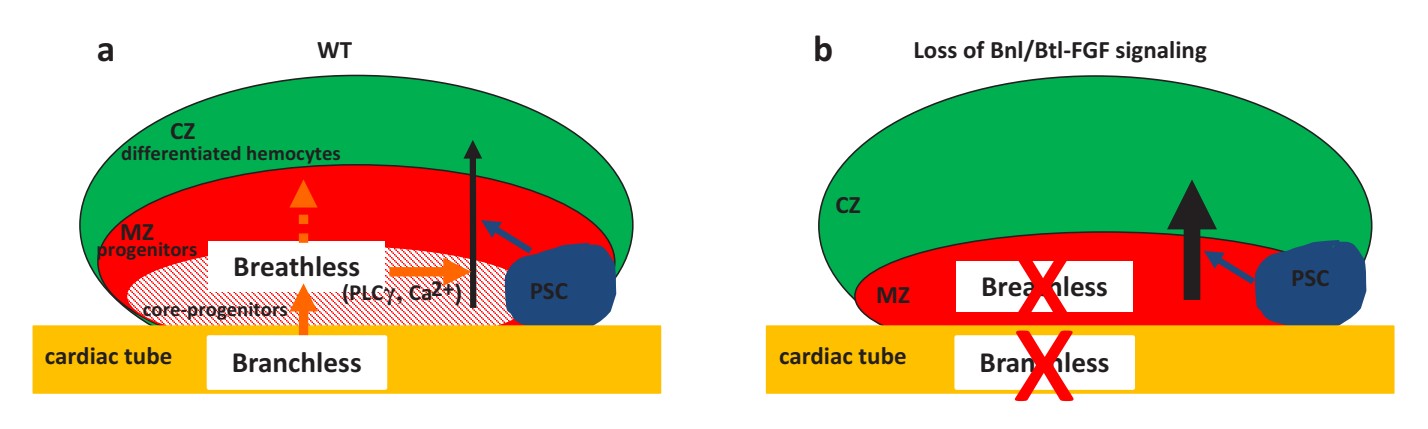

**Figure 6.** Two niches control lymph gland homeostasis. (**a–b**) Schematic representation of third instar larvae lymph gland anterior lobes. Progenitors and core progenitors are in red and hatched red, respectively. The cortical zone (CZ) is in green, the PSC and the cardiac tube (CT)/vascular system are in blue and orange, respectively. (**a**) In a wildtype (WT) lymph gland, under normal conditions the PSC, the first niche identified, regulates the maintenance of the progenitor pool except for core progenitors (blue arrow). Here, we show that by directly acting on core progenitors (orange arrow) the cardiac tube corresponds to a second niche present in the lymph gland. Bnl produced by cardiac cells activates its receptor Btl in progenitors. Btl-FGF activation regulates intracellular $Ca^{2+}$ levels via PLCγ, and controls the maintenance of core progenitors and in turn the whole progenitor pool. (**b**) When *bnl* or *btl* are knocked down in cardiac cells and progenitors, respectively, an increase in blood cell differentiation in the CZ is observed at the expense of the progenitor pool.

(*Grigorian et al., 2013*; *Krzemień et al., 2007*; *Volk et al., 2014*). The role of ECM components and associated cell-surface proteins, such as heparan sulfate proteoglycans (HSPGs) (*Muha and Müller, 2013*), in lymph gland Btl-FGF activation deserves additional investigation.

Bnl::GFP secreted by cardiac cells is detected in MZ progenitors as cytoplasmic punctate dots positive for Rab11, a marker for recycling vesicles, for Rab7, a marker for late endosomes, and for the receptor Btl. These data suggest that Bnl secreted by cardiac cells is internalized by MZ progenitors most likely through receptor-mediated endocytosis. *bnl* is transcribed in MZ progenitors and Bnl produced by these cells also contributes to lymph gland homeostasis. Taking into account the FGF dose-dependent response shown to operate in vertebrates (*Ameri et al., 2010*; *Iyengar et al., 2007*), several lymph gland sources of Bnl could be necessary to reach the threshold needed to fully activate the Btl-FGF pathway in lymph gland progenitors.

Interestingly, the Htl-FGF pathway is also required in MZ progenitors with a loss-of-function phenotype (*Dragojlovic-Munther and Martinez-Agosto, 2013*) opposite to that of to Btl-FGF inactivation. While Htl-FGF signaling acts through Ras and MAPK activation, we show here that Btl-FGF signaling controls intracellular calcium concentration in hematopoietic progenitors probably through PLCγ activation. By performing epistasis experiments, we further establish the absence of hierarchy between Htl-FGF and Btl-FGF signaling pathways in the MZ and that both pathways are required simultaneously to control lymph gland hematopoiesis. To our knowledge, this is the first example in which Btl and Htl are both expressed and required in the same cell population. We postulate that simultaneous regulation by the two pathways and a Bnl contribution by two separate tissues confers robustness to lymph gland hematopoiesis under normal developmental conditions and flexibility in response to environmental fluctuation. Since Htl and Btl inactivation leads to opposite lymph gland phenotypes, this raises the question of their respective downstream targets.

In vertebrates, FGF signaling controls both primitive and definitive hematopoiesis (*Dzierzak and Bigas, 2018*; *Muha and Müller, 2013*; *Ornitz and Itoh, 2015*). Additional studies indicate that FGFR1 in adult HSPCs is activated during hematopoietic recovery following injury, in order to stimulate HSPC proliferation and mobilization (*Zhao et al., 2012*). Furthermore, FGF2 facilitates HSPC expansion by amplifying mesenchymal stem cells, a niche cell type (*Itkin et al., 2013*; *Itkin et al., 2012*). Thus, in vertebrate adult bone marrow, the FGF pathway plays a major role in the control of hematopoiesis both under steady state conditions and in response to an immune stress. However, deciphering how FGF controls hematopoiesis in bone marrow remains an arduous task since many FGF ligands and receptors are expressed in HSPC and/or niche cells and redundancy and

compensation mechanisms between different FGF members can occur (*Haas et al., 2018*). Given the high conservation of signaling pathways between Drosophila and mammals, the low genetic redundancy in Drosophila and the striking similarities between mammalian bone marrow and fly lymph gland, there is promise that our newly identified regulation of FGF signaling in the lymph gland will shed light on the complex regulation of FGF signaling in mammalian bone marrow.

## Materials and methods

### Fly strains

 *w^{1118}* (wild type, *WT*), *UAS-mCD8-GFP* and PG125*dome-gal4* (*Krzemień et al., 2007*), *antp-gal4* (*Mandal et al., 2007*), *hand∆-gal4* (*Morin-Poulard et al., 2016*) and *NP1029-gal4* (*Monier et al., 2005*). *hand∆-gal4* corresponds to the 3^{rd} intron of *hand* deleted from the specific visceral mesoderm enhancer (*Popichenko et al., 2007* and Laurent Perrin personal communication). Lymph gland mcd8-GFP expression patterns under *hand∆-gal4* and *NP1029-gal4 drivers* in L1, L2, and L3 larvae are given in *Figure 1—figure supplement 1*. The *hand∆-gal4* is expressed in all cardiac cells, whereas the NP1029-*gal4* is expressed in all cardiac cells except those that are expressing *seven-up* (*Monier et al., 2005*). Strains used are BcGFP (*Tokusumi et al., 2009*), *bnl^{P2}* (*Sutherland et al., 1996*), hhF4-GFP (*Tokusumi et al., 2010*), domeMESO-LacZ (*Krzemień et al., 2007*), domeMESO-Gal4 (*Louradour et al., 2017*), *UAS-Bnl::GFP* (*Lin, 2009*), *UAS-btl^{CA}* on II or III (*Parés and Ricardo, 2016*), *UAS-Bnl* (*Jarecki et al., 1999*). Ubiquitin-rab11cherryFP (Y. Bellaiche), *bnl:GFP^{endo}* and *btl:cherry^{endo}* knock-in alleles (*Du et al., 2018*). Other strains were provided by the Bloomington (BL) and the Vienna (VDRC) Drosophila RNAi stock centers: *btl^{dev1}* (BL4912), *Sar1-RNAi* (BL 32364, *Cook et al., 2017*), *UAS-CaMKII* (BL 29662), *UAS-GCaMP3* (BL32116), *UAS-IP3R* (BL30741), *sl-RNAi* (BL32385 and BL35604), *sl^2* (BL724). The list of RNAi lines used for the functional screen is given in *Figure 1—figure supplement 1*. For RNAi treatments, *UAS-Dicer two* was introduced and at least two RNAi lines per gene were tested. Controls correspond to Gal4 drivers crossed with *w^{1118}*. In all experiments, crosses and subsequent raising of larvae until late L1/early L2 stage were performed at 22°C, before shifting larvae to 29°C until their dissection at the L3 stage. For gal*80^{ts}* experiments, crosses were initially maintained at 18°C (permissive temperature) for 3 days after egg laying, and then shifted to 29°C until dissection.

### RNAi screen

Antenapedia (Antp) immunostaining was revealed with the ABC kit from Abcam. The images were collected with a Nikon epifluorescence microscope. PSC cell numbers were counted manually using Fiji multi-point tool software. The BcGFP and anterior lobe areas were measured. Crystal cell index corresponds to BcGFP area/anterior lobe area. 2 RNAi lines per gene were tested when available, and at least 15 lymph glands were analyzed per genotype.

### Generation of DomeMESO-RFP transgenic lines

The domeMESO sequence from *pCasHs43domeMESO-lacZ* (*Rivas et al., 2008*) was sub-cloned into pENTR Directional, following the experimental procedure of the TOPOCloning Kit from Invitrogen. The resulting plasmid was used to generate *domeMESO-RFP* transgenic flies using attP/attB technology (*Bischof et al., 2007*). The Drosophila line was created by integration at *attP*-68A4 (III) sites.

### Antibodies and immunostaining

Lymph glands were dissected and processed as previously described (*Krzemień et al., 2007*). Antibodies used were mouse anti-Col (1/100) (*Krzemień et al., 2007*), chicken anti-βgal (1/1000, Abcam), rabbit anti-RFP (1/40 000, Rockland Immunochemicals), chicken anti-GFP(1/500, Abcam), mouse anti-Antp (1/100, Hybridoma Bank), mouse anti-Hnt (1/100, Hybridoma Bank); mouse anti-P1 (1/30, I. Ando, Institute of Genetics, Biological Research Center of the Hungarian Academy of Science, Szeged, Hungary), mouse anti-proPO (1/100, T.Trenczel, Justus-Liebig-University Giessen, Giessen, Germany). Secondary antibodies were Alexa Fluor-488 and −555 conjugated antibodies (1:1000, Molecular Probes) and goat anti-Chicken Alexa Fluor-488 (1/800; Molecular Probes). Nuclei were labeled with TOPRO3 (Thermo Fisher Scientific). Immunostainings were performed as

previously described (*Louradour et al., 2017*). For detecting bnl:GFP<sup>endo</sup> and btl:cherry<sup>endo</sup> immu-nostainings were performed with anti-GFP and anti-RFP, respectively.

## In situ hybridization

The protocol was as described in *Oyallon et al., 2016*. For fluorescent in situ hybridization we used digoxigenin-labeled *tep4* and *bnl* probes. For revelation, samples were incubated with sheep-anti-DIG (1/1000, Roche) followed by biotinylated donkey-anti-sheep (1/500, Roche). ABC kit from Vector Laboratory was used followed by fluorescent tyramide staining (Alexa fluor 555 or 488 conjugated tyramide from Molecular Probes). The *bnl* probe was transcribed in vitro using T7 RNA polymerase II, from PCR-amplified DNA sequences. Pairs of primers were used and the sequence in italics corresponds to the T7 RNA-Pol II fixation site. For *bnl*: primer 1: GCCATGGACAACAACTTGAC/*ATGAA TTCTAATACGACTCACT* ATAGGGCGTCGTTACGGTCCAGATTG; primer 2: GCAAGGCCAACAA-GAAGAAG/ATGAATTCTAATACGACTCACTATAGGGCCTGGTCGTTATCCTGATCC.

## Quantification of PSC cell numbers

In all experiments, genotypes were analyzed in parallel and quantified. PSC cells were counted manually using Fiji multi-point tool software. Statistical analyses (Mann–Whitney nonparametric test) were performed using GraphPad Prism five software.

## Blood cell and progenitor quantification

Crystal cells were visualized by either BcGFP or immunostainings with antibodies against proPO or Hnt. Plasmatocytes were labeled by P1 immunostainings. *DomeMESO-RFP* and *DomeMESO-GFP* were expressed in MZ progenitors, whereas MZ core progenitors were labeled by either *tep4* in situ hybridization or Col immunostainings. Optimized confocal sections were performed on Leica SPE or SP8 microscopes for 3D reconstruction. The numbers of crystal cells, plasmatocytes and progenitors stained and anterior lobe volume (in $\mu m^3$) were measured using Volocity 3D Image Analysis software (PerkinElmer). Crystal cell index: (crystal cell number/anterior lobe volume)x100000; plasmatocyte and progenitor index: (plasmatocyte or progenitor volume/anterior lobe volume)x100. At least 15 anterior lobes were scored per genotype, and experiments were reproduced at least three times. Statistical analyses (Mann–Whitney nonparametric test) were performed using GraphPad Prism five software. Since the number of lymph gland differentiated blood cells fluctuates depending on the larval stage, and to limit discrepancies in all the experiments, genotypes were analyzed in parallel.

## Quantification of hhF4-GFP and UAS-GCaMP3 intensity

Optimized confocal sections were performed on Leica SPE or SP8 microscopes for 3D reconstruction. For hhF4-GFP, the sum intensities for GFP per PSC labeled by Col and each PSC volume (in $\mu m^3$) were measured using Volocity 3D Image Analysis software (PerkinElmer). The intensity of hhF4-GFP corresponds to the sum intensity of hhF4-GFP/the PSC volume. For GCaMP3, the sum intensities for GFP per lymph gland primary lobe labeled by TOPRO and each primary lobe volume (in µm3) were measured using Volocity 3D Image Analysis software. The intensity of GCaMP3 corresponds to the sum intensity of GCaMP3/per lymph gland primary lobe volume. At least 15 anterior lobes were scored per genotype, and experiments were reproduced at least three times. Statistical analyses (Mann–Whitney nonparametric test) were performed using GraphPad Prism five software.

## Quantification of the diffusion in the MZ of cytoplasmic Bnl::GFP dots, in hand >Bnl::GFP and hand >Bnl::GFP >sar1 RNAi genetic contexts

Optimized lymph gland confocal sections were obtained with a Leica SP8 microscope for 3D reconstruction. The maximum projection of 10 slices chosen in the middle of the stack was performed. A parallelepiped with a larger corresponding to four nuclei diameter, a width of 10 confocal slices a length corresponding to the distance from the CT to the CZ was designed. Along the length, the parallelepiped was subdivided into 11 sub parallelepipeds of similar size. The number of Bnl::GFP granules per sub parallelepiped (called interval in *Figure 4g* legend) was counted. Spot detector plugin from ICY software (http://icy.bioimageanalysis.org/) was used to quantify the number of Bnl::GFP dots per sub parallelepiped.

## Sample size

n corresponds to the number of anterior lobes analyzed. *Figure 1*: In e, for *hand∆>* n = 61 and n = 26 for *hand∆>ilp6* RNAi. In h, for *hand∆>* n = 61 and n = 24 for *handD >ds* RNAi. In k, *hand∆>* n = 61 and n = 24 for *hand∆>pvf3* RNAi. In f, for *hand∆>* n = 19 and n = 27 for *hand∆>ilp6* RNAi. In i, for *hand∆>* n = 26 and n = 21 *for hand∆>ds* RNAi. In l, *hand∆>* n = 26 and n = 26 for *hand∆>pvf3* RNAi. In o, for *hand∆>* n = 14 and n = 10 for *hand∆>pvf3* RNAi. *Figure 2*: in d, n = 12 for WT and n = 10 for *bnl$^{P2}$/+*. In i, *hand∆>* n = 22, *hand∆>bnl*-RNAi n = 13, *Hand∆>bnl* n = 38 and *hand∆>bnl; bnl-RNAi* n = 10. In l, for *hand∆>* n = 24 and n = 15 for hand∆>bnl-RNAi. In o, for *hand∆>*n = 25 and n = 28 in *hand∆>bnl*-RNAi. For r, *hand∆>* n = 36 and n = 22 for *hand∆>bnl*-RNAi. In u, for hand∆>n = 16 and n = 18 in *hand∆>bnl*-RNAi.

*Figure 3a-a''* *domeMESO-GFP* crossed *with btl:mcherry$^{endo}$*; 3b and k: *PG125dome-gal4,UAS-dcr2* crossed with *w$^{1118}$*; c *btl$^{dev1}$/TM6B* crossed with *w$^{1118}$*; e and i: *PG125dome-gal4,UAS-dcr2* crossed with *UAS-btl-RNAi*; f: *PG125dome-gal4,UAS-dcr2* crossed with *UAS-btl$^{CA}$*; *UAS-btl-RNAi*; h: *PG125dome-gal4,UAS-dcr2; DomeMESO-LacZ* crossed with *w$^{1118}$*; i: *PG125dome-gal4,UAS-dcr2; DomeMESO-LacZ* crossed with *UAS-btl-RNAi*; n: *PG125dome-gal4,UAS-dcr2* crossed with *UAS-htl$^{DN}$*; o: *PG125dome-gal4,UAS-dcr2* crossed with *UAS-htl$^{DN}$*; *UAS-btl-RNAi*.

In d, n = 45 for *WT* and n = 30 for *btl $^{dev1}$/+*. In g, dome >n = 93, *dome >btl*-RNAi n = 23, *dome >btl$^{CA}$* n = 12 and *dome >btl$^{CA}$;btl-RNAi* n = 20. In j, n = 27 for *dome>* and n = 21 for *dome >btl*-RNAi. In m, n = 20 for *dome>* and n = 24 for *dome >btl*-RNAi. In p, n = 16 for *dome>*, n = 13 for *dome >htl$^{DN}$*. n = 18 for *dome >btl*-RNAi and n = 29 for *dome >dome > htl$^{DN}$>btl*-RNAi.

*Figure 4a* *hand∆,UAS-dcr2* crossed with *UAS-bnl::GFP*; 4b: *hand∆,UAS-dcr2; btl:mcherry$^{endo}$* crossed with *UAS-Bnl::GFP*; 4 c: *hand∆,UAS-dcr2; UAS-bnl::GFP* crossed with *ubi-rab11::mcherry*; 4d: *hand∆,UAS-dcr2* crossed with *UAS-bnl::GFP*; 4e: *hand∆,UAS-dcr2; UAS-Bnl::GFP* crossed with *w$^{1118}$*; 4 f: *hand∆,UAS-dcr2; Bnl::GFP* crossed with *UAS-sar1-RNAi*; 4 hr: *hand∆,UAS-dcr2; BcGFP* crossed with *w$^{1118}$*; 4i: *hand∆,UAS-dcr2; BcGFP* crossed with:*UAS-sar1-RNAi*; 4 j: *hand∆,UAS-dcr2; BcGFP* crossed with *UAS-sar1-RNAi; UAS-bnl::GFP*.

In k: *hand∆>* n = 27, *hand∆>sar1* RNAi n = 14, and *hand∆>sar1 RNAi; Bnl//GFP* n = 17.

*Figure 5a* *PG125dome-gal4,UAS-dcr2* crossed with *UAS-GCaMP3* ; 5b : *PG125dome-gal4,UAS-dcr2* crossed with *UAS-GCaMP3* ; *UAS-btl-RNAi* ; 5d : *PG125dome-gal4,UAS-dcr2* crossed with *w$^{1118}$*; 5e :*PG125dome-gal4,UAS-dcr2* crossed with *UAS-btl-RNAi* ; 5 f :*PG125dome-gal4,UAS-dcr2* crossed with *UAS-CaMKII* ; 5 g :*PG125dome-gal4,UAS-dcr2* crossed with *UAS-CaMKII* ; *UAS-btl-RNAi* ; 5i :*PG125dome-gal4,UAS-dcr2* crossed with *UAS-sl-RNAi* ; 5 j : *sl$^2$* ; 5 k : *sl$^2$; dome-gal4* crossed with *sl$^2$; UAS-btl-*CA .

In c, n = 41 for *dome>* and n = 29 for *dome >btl*-RNAi. In h, for *dome>* n = 35, n = 33 for *dome >btl*-RNAi, n = 35 for *dome >CaMKII* and n = 26 for *dome >CaMKII; btl-RNAi*. In l, *dome>* n = 27 and n = 30 in *dome >sl*-RNAi. In m, *WT* n = 20 and n = 17 in *sl$^2$*. In n, *sl$^2$, dome >*n = 20 and n = 21 in *sl$^2$; dome >btlCA*.

## Replicates

*Figure 2a-a''* *domeMESO-GFP* crossed with *w$^{1118}$*; *Figure 2b*: *w$^{1118}$*; *Figure 2c*: *bnl$^{P2}$/TM6B* crossed with *w$^{1118}$*; *Figure 2e*: *hand∆,UAS-dcr2; BcGFP* crossed with *w$^{1118}$*; 2 f: *hand∆,UAS-dcr2; BcGFP* crossed with *UAS-bnl-RNAi*; 2 g: *hand∆,UAS-dcr2; BcGFP* crossed with *UAS-bnl*; 2 hr: *hand∆,UAS-dcr2; BcGFP* crossed with *UAS-bnl;UAS-bnl-RNAi*; 2 j: *hand∆,UAS-dcr2; domeMESO-RFP* crossed with *w$^{1118}$*; 2 k: *hand∆,UAS-dcr2; domeMESO-RFP* crossed with *UAS-bnl-RNAi*; 2 m and p: *hand∆,UAS-dcr2* crossed with *w$^{1118}$*; 2 n and q: *hand∆,UAS-dcr2* crossed with *UAS-bnl-RNAi*. s: *hand∆,UAS-dcr2; bnl:GFP$^{endo}$* crossed with *w$^{1118}$*; t: *hand∆,UAS-dcr2; bnl:GFP$^{endo}$* crossed with *UAS-bnl-RNAi*.

(d, i) three independent experiments were performed and quantified. One is shown. (l) two independent experiments were performed and quantified. One is shown. (o) three independent experiments were performed and quantified. One is shown. (r) two independent experiments were performed and quantified. One is shown. (u) two independent experiments were performed and quantified. One is shown. *Figure 3*: (d, g) three independent experiments were performed and quantified. One is shown. (j) two independent experiments were performed and quantified. One is shown. (m) three independent experiments were performed and quantified. One is shown. (p) two independent experiments were performed and quantified. One is shown *Figure 4*: (k) two

independent experiments were performed and quantified. One is shown. (*Figure 5c,h,l–n*) two independent experiments were performed and quantified. One is shown.

## Drosophila genetics

Fly crosses for each figure:

*Figure 1c-c', m* hand∆,UAS-dcr2; BcGFP crossed with w^1118^; *Figure 1d,d'* : hand∆,UAS-dcr2; BcGFP crossed with UAS-ilp6-RNAi; *Figure 1g,g'* : hand∆,UAS-dcr2; BcGFP crossed with UAS-ds-RNAi; *Figure 1j,j', n* : hand∆,UAS-dcr2; BcGFP crossed with UAS-pvf3-RNAi.

*Figure 1—figure supplement 1a-c* hand∆,UAS-dcr2 crossed with UAS-mcd8GFP; d-f:NP1029, UAS-dcr2 crossed with UAS-mcd8GFP.

*Figure 2—figure supplement 2a* PG125dome-gal4,UAS-dcr2 crossed with w^1118^; b: PG125dome-gal4,UAS-dcr2 crossed with UAS-bnl-RNAi; d: PG125dome-gal4,UAS-dcr2 crossed with w^1118^; e: PG125dome-gal4,UAS-dcr2 crossed with UAS-bnl-RNAi; g: hand∆,UAS-dcr2 crossed with w^1118^; h: hand∆,UAS-dcr2 crossed with UAS-bnl; UAS-bnl-RNAi; j: hand∆,UAS-dcr2 crossed with w^1118^; k: hand∆,UAS-dcr2 crossed with UAS-bnl-RNAi; m: hhF4-GFP; hand∆,UAS-dcr2 crossed with w^1118^; n: hhF4-GFP; hand∆,UAS-dcr2 crossed with UAS-bnl-RNAi.

*Figure 3—figure supplement 1* a : BcGFP crossed with btl:cherry^endo^; b : hml-gal4, UAS-mcd8-GFP crossed with btl:cherry^endo^; c : pcol-gal4, UAS-mcd8-GFP crossed with btl:cherry^endo^; d : hand∆, UAS-dcr2; BcGFP crossed with w^1118^; e: hand∆,UAS-dcr2; BcGFP crossed with UAS-btl-RNAi ; g: hand∆,UAS-dcr2 crossed with w^1118^; h: hand∆,UAS-dcr2 crossed with UAS-btl-RNAi; j and m: PG125dome-gal4,UAS-dcr2 crossed with w^1118^; k and n: PG125dome-gal4,UAS-dcr2 crossed with UAS-btl-RNAi.

*Figure 4—figure supplement 1a* hand∆, UAS-dcr2 crossed with w^1118^; b: hand∆, UAS-dcr2 crossed with UAS-sar1-RNAi; d and g: NP1029, UAS-dcr2 crossed with w^1118^; e and h: hand∆, UAS-dcr2 crossed with UAS-sar1-RNAi.

*Figure 5—figure supplement 1* : a : PG125dome-gal4,UAS-dcr2 crossed with UAS-GCaMP3 ; b: PG125dome-gal4,UAS-dcr2 crossed with UAS-GCaMP3 ;UAS-btl-RNAi ; d : tep4-gal4,UAS-dcr2 crossed with w^1118^ ; e : tep4-gal4,UAS-dcr2 crossed with UAS-btl-RNAi ; f : tep4-gal4,UAS-dcr2 crossed with UAS-IP3R; g : tep4-gal4,UAS-dcr2 crossed with UAS-IP3R ; UAS-btl-RNAi.

## Sample size

*Figure 1—figure supplement 1* At least 10 anterior lobes for each condition were analyzed. *Figure 2—figure supplement 1*: (h) For hand∆ n = 16 and n = 17 for hand∆>bnl-RNAi. (k) for hand∆>n = 24 and n = 23 for hand∆>bnl-RNAi. (n) for NP1029 >n = 46 and n = 29 for NP1029 >bnl-RNAi. (q) for hand∆>; tub80ts n = 20 and n = 14 for hand∆>bnl-RNAi; tub80ts. *Figure 2—figure supplement 2*: (c) for dome >n = 24 and n = 26 for dome >bnl-RNAi. (f) dome >n = 23 and n = 24 in dome >bnl-RNAi. (i) for hand∆>n = 13, hand∆>bnl-RNAi n = 11, hand∆>bnl n = 11 and hand∆>bnl; bnl-RNAi n = 10. (l) for hand∆>n = 49 and n = 22 for hand∆>bnl RNAi. (o) for hand∆>n = 40 and n = 22 for hand∆>bnl-RNAi. *Figure 3—figure supplement 1*: (f) for hand∆>n = 23 and n = 17 for hand∆>btl-RNAi. (i) for hand∆>n = 50 and n = 34 for hand∆>btl-RNAi. (l) For dome >n = 26 and n = 22 for dome >btl-RNAi. (o) for dome >n = 21 and n = 23 for dome >btl-RNAi. *Figure 4—figure supplement 1*: (c) for hand∆>n = 37 and n = 28 for hand∆>sar1 RNAi. (f) for NP1029 >n = 20 and n = 28 for NP1029 >sar1 RNAi. (i) for NP1029 >n = 15 and n = 28 for NP1029 >sar1 RNAi. *Figure 5—figure supplement 1*: (c) for dome >n = 14 and for dome >btl-RNAi n = 7. (h) for tep4 >n = 14, for tep4 >btl-RNAi n = 12, for tep4 >IP3R n = 27 and for tep4 >IP3R; btl-RNAi n = 27.

## Replicates

*Figure 2—figure supplement 1a* BcGFP crossed with w^1118^; b: hml-gal4, UAS-mcd8GFP crossed with w^1118^; c: pcol-gal4, UAS-mcd8-GFP crossed with w^1118^; d: hand∆,UAS-dcr2 crossed with UAS-mcd8-GFP; e: hand∆,UAS-dcr2 crossed UAS-mcd8-GFP; UAS-bnl-RNAi; f: hand∆,UAS-dcr2 crossed with w^1118^; g: hand∆,UAS-dcr2 crossed with UAS-bnl-RNAi; i: hand∆,UAS-dcr2 crossed with w^1118^; j: hand∆,UAS-dcr2 crossed with UAS-bnl-RNAi 34572; l: NP1029, UAS-dcr2 crossed with w^1118^; m: NP1029, UAS-dcr2 crossed with UAS-bnl-RNAi; o: hand∆,UAS-dcr2;tub- gal80^ts^ crossed with w^1118^; p: hand∆,UAS-dcr2;tub- gal80^ts^ crossed with UAS-bnl-RNAi.

(h)two independent experiments were performed and quantified. One is shown. (k and n) three independent experiments were performed and quantified. One is shown. (q) two independent experiments were performed and quantified. *Figure 2—figure supplement 2*: (c) three independent experiments were performed and quantified. One is shown. (f and i) two independent experiments were performed and quantified. One is shown. (l and o) three independent experiments were performed and quantified. One is shown. *Figure 3—figure supplement 1*: (f) three independent experiments were performed and quantified. One is shown. (i). two independent experiments were performed and quantified. One is shown. (l and o) three independent experiments were performed and quantified. One is shown. *Figure 4—figure supplement 1*: (c, f and i) two independent experiments were performed and quantified. One is shown. *Figure 5—figure supplement 1*: (c and h) two independent experiments were performed and quantified. One is shown.

## Acknowledgements

We thank M Affolter, Y Bellaiche, J Casanova, J Colombani, L Du, M Freeman, M Grammont, M A Krasnow, A Paululat, L Perrin, S Ricardo, S Roy, T Tanaka, P Thérond, Bloomington and Vienna Stock Center and the TRiP at Harvard Medical School for fly strains; I Ando, A Moore and T Trenczek for antibodies; L Bataillé, A Davy, G Lebreton, M Meister, C Monod, B Monnier and A Vincent, for critical reading of the manuscript. We are grateful to B Ronsin and S Bosch for assistance with confocal microscopy (Plateforme TRI); J Favier, V Nicolas and A Destenable for fly culture. Research in the authors' laboratory is supported by the CNRS, University Toulouse III, Ministère de la Recherche (ANR « programme blanc »), ARC (Association pour la Recherche sur le Cancer), La Ligue contre le Cancer 31, La Société Française d'Hématologie (SFH), FRM (Fondation pour la Recherche Médicale) and the China Scholarship Council.

## Additional information

### Funding

| Funder | Grant reference number | Author |
|---|---|---|
| Ministère de l'Education Nationale, de l'Enseignement Superieur et de la Recherche | 'ANR programme blanc' | Michele Crozatier |
| Fondation ARC pour la Recherche sur le Cancer | ARC | Manon Destalminil-Letourneau |
| Fondation pour la Recherche Médicale | FRM | Michele Crozatier |
| Ligue Contre le Cancer | | Michele Crozatier |
| China Scholarship Council | | Yushun Tian |
| Société francaise d'hématologie | | Manon Destalminil-Letourneau |

The funders had no role in study design, data collection and interpretation, or the decision to submit the work for publication.

### Author contributions

Manon Destalminil-Letourneau, Conceptualization, Formal analysis, Investigation, Methodology, Writing - original draft; Ismaël Morin-Poulard, Conceptualization, Formal analysis, Investigation, Methodology, Writing - review and editing; Yushun Tian, Nathalie Vanzo, Formal analysis; Michele Crozatier, Conceptualization, Data curation, Formal analysis, Supervision, Funding acquisition, Validation, Investigation, Writing - original draft, Project administration, Writing - review and editing

### Author ORCIDs

Nathalie Vanzo http://orcid.org/0000-0002-6659-0299
Michele Crozatier https://orcid.org/0000-0001-9911-462X

Decision letter and Author response
Decision letter https://doi.org/10.7554/eLife.64672.sa1
Author response https://doi.org/10.7554/eLife.64672.sa2

## Additional files

### Supplementary files
• Transparent reporting form

### Data availability
All data generated or analysed during this study are included in the manuscript and supporting files. Source data files have been provided for all figures.

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
