## [Decision Letter]

**Acceptance summary:**

This paper provides mechanistic insights into the role of the cardiac tube as a vascular niche controlling blood progenitor maintenance and differentiation by the Bnl-Btl arm of FGF signalling via Calcium. The findings in the manuscript are impactful and will contribute an important aspect to the field of hematopoiesis.

**Decision letter after peer review:**

[Editors’ note: the authors submitted for reconsideration following the decision after peer review. What follows is the decision letter after the first round of review.]

Thank you for submitting your work entitled "The vascular niche controls *Drosophila* hematopoiesis via Fibroblast Growth Facto signaling" for consideration by *eLife*. Your article has been reviewed by three peer reviewers, one of whom is a member of our Board of Reviewing Editors, and the evaluation has been overseen by a Senior Editor. The following individual involved in review of your submission has agreed to reveal their identity: Tina Mukherjee (Reviewer #2).

Our decision has been reached after consultation between the reviewers. Based on these discussions and the individual reviews below, we regret to inform you that your work will not be considered further for publication in *eLife*.

While the reviewers agree that this paper has its merits and deserves publication, they feel that the current manuscript has too many conceptual concerns that need clarifications which are absolutely necessary to strengthen the core idea being proposed in the study. The general consensus was to reject the manuscript but allows re-submission. This provides more time than the usual 2 months to revise the manuscript. A re-submission will be considered as new submission. Taken in consideration the concerns, you might decide to submit the manuscript to another journal. However, if you decide to resubmit, we will try to send the manuscript to the same reviewers unless you instruct us otherwise.

Reviewer #1:

In this manuscript, Destalminil-Letourneau et al., describes a signaling crosstalk between the cardiac cells and the hematopoietic progenitors that regulate *Drosophila* lymph gland homeostasis independently from the PSC. The authors identified through a genetic screen the Fibroblast Growth Factor (FGF) ligand Branchless (Bnl) produced by cardiac cells that maintain hematopoietic lymph gland progenitors by controlling intracellular Ca^2+^ concentration via PLCγ activation. This result brings advancement in the study of central hematopoiesis in the *Drosophila* larvae, although, it was already shown that the cardiac cells impact central hematopoiesis via the regulation of PSC morphology. While this article has some merits, I am not sure that it is strong enough to deserve publication in e*Life*.

First, this newly identified pathway is not integrated with the other pathways that regulate lymph gland homeostasis already described by this team and other research groups. As such, it is unclear what are the specificity of this pathway and what it brings to the process of hematopoiesis. In order to answer these questions, it would be necessary to analyse the impact of the cardiac cell-progenitor loop on hematopoiesis during an infestation.

1) A first concern is that this new pathway involving cardia cells and progenitor through Bnl and Btl is not integrated with other pathways. There is only a little insight provided on how this signaling cascade impact differentiation except for an increase of Ca^2+^.

2) A second concern here is that this inter-organ communication between the cardiac cells and the lymph gland has been studied under homeostatic but not immune stress conditions. Some additional experiments are needed to show how the cardiac cells might act as a niche to regulate the hematopoietic response to immune stress such as wasp parasitism.

3) The authors demonstrated the diffusion of Bnl from the cardiac cells to the lymph gland progenitors. However, the data concerning the internalization of Bnl in the lymph gland by endocytosis is not convincing. First, Figure 4 (B,B',B') is unclear. More generally, the authors should look at the colocalization of Bnl-GFP with the early endosomal Rab5 and the late endosomal Rab7 that have a key role in the transport along the endocytic pathway. In Figure 4 (A,A'): They should use an endogenously tagged version of Btl to validate further that Bbl is secreted by the cardiac cells and not only an over-expression of Bnl tagged protein.

4) A weakness of the manuscript is that we do not know the time during larval development at which this pathway plays a role.

Reviewer #2:

In this study, the authors describe the importance of cardiac/vascular cells in regulating the hematopoietic progenitor maintenance in the lymph gland. The authors propose that FGF ligand Branchless (identified as a result of a functional screen) is emanated by the vascular cells which activates FGF receptor, Breathless in the progenitor cells, necessary to maintain their homeostasis and prevent excessive differentiation. Down-stream of Btl receptor activation, the authors demonstrate the involvement of PLC-γ mediated control of progenitor calcium levels that is necessary to execute their homeostasis. Overall, the present work attempts to address the fundamental concept of understanding unidentified signal from vascular niche to control maintenance of differential population of hematopoietic progenitor cells, via utilizing a simpler and a conserved model organism *Drosophila* with allied similarity with the vertebrate bone marrow system. Their perusal for finding additional signals from the vascular cells in regulating hematopoiesis stems from an earlier study where PSC dependent and independent progenitors were shown to reside in the medullary zone suggesting heterogeneity in the progenitor population (Baldeosingh et al., 2018).

Although the findings seem appealing and interesting, there are certain concerns that I have regarding the main conclusions drawn in the manuscript and the experimental strategies employed to infer them. I list them point-by-point below. It is important the authors address them to prove their model and demonstrate clarity in their proposed work.

a) Specificity of Bnl source: The major concern that I have in this manuscript is regarding the specificity by which the authors prove it is the cardiac cells derived Bnl whose function is necessary for progenitor homeostasis. Given that there are two sources of Bnl production, the cardiac cells and the progenitor cells themselves, the HandΔ-gal4 line utilized in this study is confusing in accurately dissecting the source. The specificity of HandΔ-gal4, which the authors claim is a cardiac cell-specific driver, is also reported to be expressed in early LG hematopoietic compartment cells (Hand is a direct target of Tinman and GATA factors during *Drosophila* cardiogenesis and hematopoiesis http://dev.biologists.org/content/132/15/3525), (Pvr expression regulators in equilibrium signal control and maintenance of *Drosophila* blood progenitors https://www.ncbi.nlm.nih.gov/pmc/articles/PMC4185420/), and lineage tracing with Hand-gal4 marks lymph gland progenitor cells. Given the progenitor differentiation phenotype observed upon blocking Bnl using Dome-gal4, the interpretation of the cardiac cells being the source is confusing and needs to be clarified. I understand that the authors have utilized NP1029-gal4, but again, if this has any overlapping expression in the lymph gland remains unaddressed. A lineage tracing of NP1029-gal4 to show no overlapping expression in the lymph gland is important. Secondly, both Hand and Dome are co-expressed very early (24hours AEL) in lymph gland progenitor cells, following which Hand is only restricted to cardiac and PSC cells, while Dome continues to be expressed in progenitor cells. If the loss of progenitor maintenance in HandΔ>BnlRNAi or Dome>BnlRNAi is a consequence of this early over-lapping expression needs to be tested. Temporal analysis of the lymph gland phenotype using gal80ts based experiments should help resolve this concern.

b) Temporal role of Bnl in progenitor cells: The two Bnl sources, cardiac and blood cells, loss of Bnl in either gives the same phenotype. It is important to address the temporal requirement of Branchless and the dependence or independence of one source over the other to highlight the importance of the cardiac niche in establishing progenitor homeostasis. In its current form, the manuscript fails to highlight the importance of this niche. A comparative analysis of Bnl-GFP expression during lymph gland development (from early to late3rd instar) should be undertaken to reveal its expression profile within the cardiac cells and blood progenitor cells. Secondly, changes in Bnl-GFP pattern upon expressing BnlRNAi in cardiac cells or using Dome-gal4 will hopefully address the important contribution of cardiac niche in regulating progenitor Bnl levels. Finally, with regards to the role of Bnl, the other conceptual concern that is raised is its requirement either as a progenitor development signal or that it is required post progenitor development only as a maintenance cue. Again, using gal80ts as mentioned in the previous comment, should help clarify this aspect.

c) Calcium homeostasis and FGF signaling: Although the authors show a down-regulation of GCAMP3 expression in lymph gland progenitor cells upon loss of FGF signaling, the analysis has been mostly done in the 3rd instar lymph gland when most of the tissue is differentiated. Hence it is hard to predict if the down-regulation is indeed because of loss of FGF signaling or is a consequence of progenitor differentiation and loss of these cells that maintain elevated GCAMP3 expression. Analysis of GCAMP3 levels in 2nd instar lymph glands prior to the onset of differentiation will help resolve this matter. Secondly, over-expression of PLC-γ in BtlRNAi condition may be important to prove the connection between Btl and activation of its downstream cascade linking to Calcium homeostasis more affirmatively.

d) Cardiac niche to maintain progenitor heterogeneity: The authors talk about heterogeneity in progenitors and the importance of the cardiac niche in this "Furthermore, the MZ progenitor population is heterogeneous and a subset of progenitors, called "core progenitors", which express […]these data led us to ask whether signals derived from cardiac cells were involved in the control of lymph gland homeostasis, i.e.: the balance between progenitors and differentiated blood cells, independently from the PSC". This idea doesn't seem to crystallize in the course of the findings made. The readouts for the assays done are looking at crystal cell differentiation and progenitor maintenance status to decipher FGF signaling in progenitors with ligand contribution from cardiac cells. There needs to be some way to reconcile this either experimentally (differential effects on Tep4 and Dome expression under some genetic manipulations already shown in the manuscript) or textually in the Discussion.

Reviewer #3:

In this manuscript, Destalminil-Letourneau et al. describe a novel mechanism of progenitor maintenance in the hematopoietic organ, the lymph gland. They provide evidence for the presence of a vascular niche via cardiac cells, that regulates blood progenitor maintenance in the lymph gland. This manuscript provides interesting and novel insights by showing that the existence of a vascular niche as a conserved mechanism of blood stem cell maintenance as it also exists in flies. The authors provide mechanistic insight by providing data that supports the assertion that vascular niche-lymph gland mediated FGF pathway (via Bnl-Btl) signalling positively regulates hematopoietic progenitor maintenance by controlling intracellular calcium levels in the medullary zone of the lymph gland. The manuscript reports some exciting findings but could benefit from a few improvements requiring further experimentation, validation, and analysis. Major comments are as follows:

1) In Figure 2A-A' the authors that the Bnl signal is present throughout the primary lymph gland lobe. This raises the possibility that cells in Posterior Signaling Centre or the Cortical Zone can also be a source of Bnl for the prohemocytes via a type of reciprocal signalling. One way to look at the possibility that the PSC cells or CZ cells are a potential source of Bnl is to include high resolution images where Bnl is detected (either via antibody or in situ) and simultaneously label PSC cells (with collier or Antp-Gal4 driven mCD8GFP) or CZ cells (for example with Hmldelta-Gal4 driven mCD8GFP) with an in-situ against Bnl or with Bnl antibody – to rule out the possibility.

2) The author's arguments about tissue specific requirement of Bnl would be strengthened by testing whether the increased differentiation in bnlP2 mutants can be rescued by restoring bnl levels by transgene-mediated expression in the cardiac tube and the MZ. For example it could be determined whether the constitutive activation of Btl in prohemocytes in the genetic background of Btl mutants used (btldev1/+) rescues the crystal cell and plasmatocyte differentiation. Additionally, the authors could check if expressing Bnl in the MZ using dome or tep4-Gal4 can rescue the BnlRNAi phenotype of increased prohemocyte differentiation.

3) Figure 3A also shows strong expression of Btl receptor in the cells towards the cortical zone (periphery of the LG). This raises some intriguing mechanistic questions. First, the manuscript would greatly benefit with a better, systematic, analysis of Bnl and Btl expression both in each of the zonal compartments with appropriate markers (PSC, MZ and CZ) – with high magnification/high resolution images. Second, One can envision a scenario where are the cells in the CZ themselves act as a source of Bnl that binds to Btl in neighboring (MZ) cells to regulate differentiation. Currently the authors have not ruled out this alternative model. If the Bnl is indeed secreted by differentiated cells then there could be 2 modes of signalling – either cell autonomous/paracrine mode or reciprocal signalling to maintain the MZ. It would be helpful if further clarification is provided of the role played by Bnl/Btl-FGF signalling in the CZ. It is possible is that the Bnl secreted by the cardiac cells is transcytosed/transported to the differentiated cells where it regulates differentiation cell autonomously. The authors could test this by perturbing Bnl and Btl in the CZ (using Hmldelta-Gal4 or eater-Gal4 for plasmatocytes and lz-Gal4 for crystal cells) and asking whether this affects crystal cell or plasmatocyte differentiation cell autonomously.

4) As noted by the authors it was previously shown that another FGF pathway in *Drosophila*, mediated by Heartless, is required for regulating hematopoiesis in the lymph gland (Dragojlovic-Munther et al., 2013). In this previous report overexpressing Btl in the progenitors had no effect (Figure S6B and K of that paper), this merits some discussion ad explanation from the authors. Furthermore, since both Btl and Heartless signalling work through the transcription effector gene pointed we would expect cross talk between the two FGF pathways. This raises an important question of how the two pathways coordinated in the lymph gland. Some genetic interaction analysis between the components of the two arms of FGF signalling will be particularly useful. Specifically, determining if one of the FGF pathways is epistatic to the other, as well as establishing whether indeed both pathways converge on Pointed.

5) The Rab11 data in Figure 4B-B' is not as strong as it could be. First, the addition of a cell membrane marker and images of a better and higher resolution would allow greater clarity. Also, since Rab11 marks the recycling endosome, it would be more appropriate to look at other endocytic markers like Rab5 (early endosome marker) and/or Rab7 (late endosome) in addition to Rab11. Also, the co-expression data in the above cases with the Rabs is currently qualitative. It is appropriate in this sort of instance to include quantitative measurement of co-localization. The authors should consider including a staining showing endocytic localization (marked with the Rabs) of the Btl receptor (by in-situ or by antibody) in the prohemocytes. Also in Figure 4B-B' shows only two cells which have Rab11-bnl co-expression/co-localization whereas the field of interest seems to have many more cells in the area. Why don't all the prohemocytes show this co-expression? Finally, also in Figure 4 Bnl diffusion data in a-a' does not look convincing without a cellular marker to identify the hemocyte populations in the LG. Such a marker should be added.

---

## [Author Response]

[Editors’ note: the authors resubmitted a revised version of the paper for consideration. What follows is the authors’ response to the first round of review.]

Reviewer #1:In this manuscript, Destalminil-Letourneau et al., describes a signaling crosstalk between the cardiac cells and the hematopoietic progenitors that regulate *Drosophila* lymph gland homeostasis independently from the PSC. The authors identified through a genetic screen the Fibroblast Growth Factor (FGF) ligand Branchless (Bnl) produced by cardiac cells that maintain hematopoietic lymph gland progenitors by controlling intracellular Ca^2+^ concentration via PLCγ activation. This result brings advancement in the study of central hematopoiesis in the *Drosophila* larvae, although, it was already shown that the cardiac cells impact central hematopoiesis via the regulation of PSC morphology. While this article has some merits, I am not sure that it is strong enough to deserve publication in eLife.

We were very surprised and disappointed by this comment, and thus we want to clarify what the novelties in this study are and their important impact for future investigations. Previous studies established that the PSC acts as a niche to control lymph gland homeostasis (for review Letourneau et al., 2016 and Banerjee et al., 2019) The hematopoietic progenitor pool in the lymph gland is heterogeneous, and a subset is not controlled by PSC signals (Baldeosingh et al., 2018). We showed previously that the cardiac/vascular system controls the PSC size and its function. Thus, through an indirect regulation involving the PSC, cardiac cells control lymph gland hematopoiesis (Morin–Poulard et al., 2016).

In this study, for the first time we provide evidence that the vascular system which directly controls blood cell progenitors independently from the PSC acts as a niche. Thus, these data establish that 2 niches control the lymph gland. Furthermore, we provide evidence that through the activation of Fibroblast Growth Factor (FGF) signaling, the vascular system prevents hematopoietic progenitors from massive differentiation, ensuring the proper balance between blood cell populations within the lymph gland. Finally, FGF activation in blood cell progenitors prevents their differentiation by regulating their intracellular calcium levels probably via PLCg activation. We sincerely think that these data bring very important and novel knowledge on the mechanisms that control lymph gland hematopoiesis and open new avenues of investigation.

First, this newly identified pathway is not integrated with the other pathways that regulate lymph gland homeostasis already described by this team and other research groups. As such, it is unclear what are the specificity of this pathway and what it brings to the process of hematopoiesis. In order to answer these questions, it would be necessary to analyse the impact of the cardiac cell-progenitor loop on hematopoiesis during an infestation.1) A first concern is that this new pathway involving cardia cells and progenitor through Bnl and Btl is not integrated with other pathways. There is only a little insight provided on how this signaling cascade impact differentiation except for an increase of Ca^2+^.

We think that establishing that Bnl/Btl activation in lymph gland progenitors acts by controlling Ca^2+^ levels through the activation of PLCg is not of little insight but rather reveals a novel yet unsuspected mechanism. Furthermore, we analyze the relationship between BtlFGF and Htl-FGF signaling that was previously reported to be required in MZ progenitors (Dragojlovic-Munther and Martinez-Agosto, 2013). We provide evidence that there is no hierarchy between Btl-FGF and Htl-FGF signaling in the MZ. Altogether, these data reveal a primary level of integration that occurs in the lymph gland.

Unraveling the integration of Bnl/Btl activation with other pathways that regulate lymph gland homeostasis represents a huge task, since many regulators (and the list is still growing) have been identified, and there is evidence for both intrinsic and extrinsic regulations (for a review please see Letourneau et al., 2016 and Banerjee et al., 2019). If the analysis is restricted to the mechanisms that control lymph gland MZ progenitors, two main problems arise. Indeed, most studies examining the role of genes/signaling pathways involved in MZ progenitors do not distinguish whether they are required for progenitor specification at the L1 or L2 larval stages, or for progenitor maintenance at the L3 stage. In addition, most of them analyze the lymph gland progenitor pool as labelled by dome>GFP without looking at defects in the “core progenitor” pool (labelled by Col and tep4). Thus, before being able to “integrate Bnl/Btl signaling”, one would have to reinvestigate the previously identified genes/signaling pathways with respect to when and in which lymph gland progenitor pools they are required. This represents as such entirely independent studies that could take years and are out of the focus of this paper.

2) A second concern here is that this inter-organ communication between the cardiac cells and the lymph gland has been studied under homeostatic but not immune stress conditions. Some additional experiments are needed to show how the cardiac cells might act as a niche to regulate the hematopoietic response to immune stress such as wasp parasitism.

We performed wasp parasitism when bnl (hand>bnl-RNAi) or btl (dome>btl-RNAi) were knocked down in cardiac cells and progenitors respectively, and measured the % of wasp egg encapsulation. In both conditions, a defect in wasp egg encapsulation was observed compared to control, indicating that Bnl from cardiac cells and Btl in progenitors are required for an efficient cellular immune response against wasps. But now a detailed analysis is required to decipher how Btl-FGF signaling is involved. This question is out of the scope of this paper and will be the focus of intense future analyses.

3) The authors demonstrated the diffusion of Bnl from the cardiac cells to the lymph gland progenitors. However, the data concerning the internalization of Bnl in the lymph gland by endocytosis is not convincing. First, Figure 4 (B,B',B') is unclear. More generally, the authors should look at the colocalization of Bnl-GFP with the early endosomal Rab5 and the late endosomal Rab7 that have a key role in the transport along the endocytic pathway. In Figure 4 (A,A'): They should use an endogenously tagged version of Btl to validate further that Bbl is secreted by the cardiac cells and not only an over-expression of Bnl tagged protein.

In agreement with the reviewer’s comment we better defined bnl::GFP propagation from cardiac cells to MZ progenitors (HandD>Uas bnl::GFP). We used the ubi-Rab11-cherryFP reporter and performed Rab7 immunostaining, which label recycling vesicles and late endosomes, respectively. Unfortunately we cannot look at Rab5 vesicles since the Rab5 reporter is tagged by GFP and therefore colocalisation with Bnl::GFP cannot be done. We also performed Col immunostaining to visualize core progenitors. These additional data are provided in Figure 4 A, C-D”. In addition, we looked at Bnl::GFP colocalisation with the Btl receptor in progenitors (Figure 4 B-B”’) and finally at Bnl::GFP diffusion from the CT to the CZ when the secretion of cardiac cells was impaired by *sar1-RNAi* expression. These data are provided in Figure 4 E-G. Altogether, they strongly support the proposition that Bnl::GFP secreted by cardiac cells is internalized in MZ progenitors likely through receptor-mediated endocytosis. This is discussed in the revised version of the manuscript.

Finally we analyzed endogenous Bnl expression using the *bnl:GFP^endo^* knock-in allele (Du et al., 2018). When *bnl* was knocked-down in the endo cardiac tube (*handD>bnl RNAi, bnl:GFP^endo^*) lower Bnl:GFP^endo^ levels were recorded in MZ progenitors (Figure 2S-U). This result illustrates that Bnl secreted by cardiac cells contributes to Bnl levels in progenitors. These new data have been added in the figures and in the text.

4) A weakness of the manuscript is that we do not know the time during larval development at which this pathway plays a role.

Sorry for not being clear enough in the previous version of the manuscript. In all experiments, crosses and subsequent raising of larvae until late L1/early L2 stage were performed at 22°C, before shifting larvae to 29°C until their dissection at the L3 stage. This information is now given both in the Materials and methods and the Results sections. Furthermore, according to the reviewer’s advice we performed temporal analysis of the lymph gland phenotype using tub-gal80^ts^ (Figure 2—figure supplement O-Q) and looked at GCaMP3-expression at the L2 stage (Figure 5—figure supplement 1A-C). In conclusion, bnl/btl-FGF signaling is not involved in progenitor development till L2 but is required in third instar larvae to control the balance between MZ progenitor maintenance and hemocyte differentiation. This is now clearly stated in the revised manuscript.

Reviewer #2:In this study, the authors describe the importance of cardiac/vascular cells in regulating the hematopoietic progenitor maintenance in the lymph gland. The authors propose that FGF ligand Branchless (identified as a result of a functional screen) is emanated by the vascular cells which activates FGF receptor, Breathless in the progenitor cells, necessary to maintain their homeostasis and prevent excessive differentiation. Down-stream of Btl receptor activation, the authors demonstrate the involvement of PLC-γ mediated control of progenitor calcium levels that is necessary to execute their homeostasis. Overall, the present work attempts to address the fundamental concept of understanding unidentified signal from vascular niche to control maintenance of differential population of hematopoietic progenitor cells, via utilizing a simpler and a conserved model organism *Drosophila* with allied similarity with the vertebrate bone marrow system. Their perusal for finding additional signals from the vascular cells in regulating hematopoiesis stems from an earlier study where PSC dependent and independent progenitors were shown to reside in the medullary zone suggesting heterogeneity in the progenitor population (Baldeosingh et al., 2018).Although the findings seem appealing and interesting, there are certain concerns that I have regarding the main conclusions drawn in the manuscript and the experimental strategies employed to infer them. I list them point-by-point below. It is important the authors address them to prove their model and demonstrate clarity in their proposed work.a) Specificity of Bnl source: The major concern that I have in this manuscript is regarding the specificity by which the authors prove it is the cardiac cells derived Bnl whose function is necessary for progenitor homeostasis. Given that there are two sources of Bnl production, the cardiac cells and the progenitor cells themselves, the HandΔ-gal4 line utilized in this study is confusing in accurately dissecting the source. The specificity of HandΔ-gal4, which the authors claim is a cardiac cell-specific driver, is also reported to be expressed in early LG hematopoietic compartment cells (Hand is a direct target of Tinman and GATA factors during *Drosophila* cardiogenesis and hematopoiesis http://dev.biologists.org/content/132/15/3525), (Pvr expression regulators in equilibrium signal control and maintenance of *Drosophila* blood progenitors https://www.ncbi.nlm.nih.gov/pmc/articles/PMC4185420/), and lineage tracing with Hand-gal4 marks lymph gland progenitor cells. Given the progenitor differentiation phenotype observed upon blocking Bnl using Dome-gal4, the interpretation of the cardiac cells being the source is confusing and needs to be clarified. I understand that the authors have utilized NP1029-gal4, but again, if this has any overlapping expression in the lymph gland remains unaddressed. A lineage tracing of NP1029-gal4 to show no overlapping expression in the lymph gland is important. Secondly, both Hand and Dome are co-expressed very early (24hours AEL) in lymph gland progenitor cells, following which Hand is only restricted to cardiac and PSC cells, while Dome continues to be expressed in progenitor cells. If the loss of progenitor maintenance in HandΔ>BnlRNAi or Dome>BnlRNAi is a consequence of this early over-lapping expression needs to be tested. Temporal analysis of the lymph gland phenotype using gal80ts based experiments should help resolve this concern.

We apologize for creating confusion about the HandD-gal4 driver we used. The HandD-gal4 diver is NOT the hand-gal4 driver (also called hand cardiac and hematopoiesis (HCH)-gal4) described in (Han and Olson, 2005) and used in the 2 papers cited by the reviewer. The HandD-gal4 has been used previously in Morin–Poulard et al., 2016. In larvae, the HandD-gal4 driver is expressed in cardiac cells and pericardial cells but not in lymph gland cells. The expression profiles of HandD-gal4>GFP and NP1029-gal4>GFP in L1, L2 and L3 larval lymph glands are now provided in Figure 1—figure supplement 1 A-F’. Complementary information concerning these drivers is added in Materials and methods.

In all experiments, crosses and the subsequent raising of larvae until late L1/early L2 stage were performed at 22°C, before shifting larvae to 29°C until their dissection at the L3 stage. This information is now given both in the Materials and methods and the Results sections. Furthermore, according to the reviewer’s advice we performed temporal analysis of the lymph gland phenotype using tub-gal80ts (Figure 2—figure supplement 1O-Q) and looked at GCaMP3-expression at the L2 stage (Figure 5—figure supplement 1A-C). In conclusion, bnl/btl-FGF signaling is not involved in progenitor development till L2 but is required in third instar larvae to control the balance between MZ progenitor maintenance and hemocyte differentiation. This is now clearly stated in the revised manuscript.

b) Temporal role of Bnl in progenitor cells: The two Bnl sources, cardiac and blood cells, loss of Bnl in either gives the same phenotype. It is important to address the temporal requirement of Branchless and the dependence or independence of one source over the other to highlight the importance of the cardiac niche in establishing progenitor homeostasis. In its current form, the manuscript fails to highlight the importance of this niche. A comparative analysis of Bnl-GFP expression during lymph gland development (from early to late3rd instar) should be undertaken to reveal its expression profile within the cardiac cells and blood progenitor cells. Secondly, changes in Bnl-GFP pattern upon expressing BnlRNAi in cardiac cells or using Dome-gal4 will hopefully address the important contribution of cardiac niche in regulating progenitor Bnl levels. Finally, with regards to the role of Bnl, the other conceptual concern that is raised is its requirement either as a progenitor development signal or that it is required post progenitor development only as a maintenance cue. Again, using gal80ts as mentioned in the previous comment, should help clarify this aspect.

A detailed expression profile of Bnl is now provided. Since Bnl is diffusible, we performed ISH with simultaneous immunostainings with markers for different lymph gland cell types. These new data are provided in Figure 2a-a” and in Figure 2—figure supplement 1A-C”. We also looked at endogenous Bnl protein using the bnl:GFP^endo^ knock-in alleles described by Du et al., 2018 (please see Figure 2S-T’).

When *bnl* was knocked-down in the cardiac tube (*handD>bnl RNAi, bnl:GFP^endo^* ) lower Bnl:GFP*^endo^* levels were recorded in MZ progenitors (Figure 2S-U). This result illustrates that Bnl secreted by cardiac cells contributes to Bnl levels in progenitors. These novel data are provided in Figure 2 and discussed in the text.

Concerning the temporal requirement of Bnl/Btll-FGF signaling, please see above our answer to point a.

c) Calcium homeostasis and FGF signaling: Although the authors show a down-regulation of GCAMP3 expression in lymph gland progenitor cells upon loss of FGF signaling, the analysis has been mostly done in the 3rd instar lymph gland when most of the tissue is differentiated. Hence it is hard to predict if the down-regulation is indeed because of loss of FGF signaling or is a consequence of progenitor differentiation and loss of these cells that maintain elevated GCAMP3 expression. Analysis of GCAMP3 levels in 2nd instar lymph glands prior to the onset of differentiation will help resolve this matter. Secondly, over-expression of PLC-γ in BtlRNAi condition may be important to prove the connection between Btl and activation of its downstream cascade linking to Calcium homeostasis more affirmatively.

As proposed by the reviewer, we looked at lymph gland GCaMP3 levels in second instar larvae. No difference compared to controls was observed (Figure 5—figure supplement 1 A-C), indicating that there is no difference in MZ progenitors till L2 stage.

Thanks to the reviewer’s remark about the connection between Btl and PLC-, we examined the relationship between *sl* and the Btl-FGF pathway and performed epistasis experiments (Figure 5J, K-N). Our data establish that *sl* acts downstream of the Bnl/Btl-FGF pathway.

Altogether, these data support the conclusion that in MZ progenitors, Bnl/Btl-FGF signaling leads to the activation of PLCg, which controls Ca^2+^ levels and in turn hemocyte differentiation. These new data are provided in (Figure 5J, K-N), discussed in the text and indicated on the model in Figure 6.

d) Cardiac niche to maintain progenitor heterogeneity: The authors talk about heterogeneity in progenitors and the importance of the cardiac niche in this "Furthermore, the MZ progenitor population is heterogeneous and a subset of progenitors, called "core progenitors", which express […] these data led us to ask whether signals derived from cardiac cells were involved in the control of lymph gland homeostasis, i.e.: the balance between progenitors and differentiated blood cells, independently from the PSC". This idea doesn't seem to crystallize in the course of the findings made. The readouts for the assays done are looking at crystal cell differentiation and progenitor maintenance status to decipher FGF signaling in progenitors with ligand contribution from cardiac cells. There needs to be some way to reconcile this either experimentally (differential effects on Tep4 and Dome expression under some genetic manipulations already shown in the manuscript) or textually in the Discussion.

Our apologies for not being clear on this point. It has been previously established that lymph gland MZ progenitors are labelled by either dome-MESO-LacZ (Krzemien et al., 2007) or domeMESO-RFP (Louradour et al., 2017). Col and *tep4* are expressed in a subset of MZ progenitors called “the core progenitors”, which also express domeMESO-LacZ or domeMESO-RFP (Oyallon et al., 2016). In this study we carefully distinguished both progenitor types using the corresponding markers (DomeMESO-RFP and *tep4* and/or Col). Knocking down *bnl* in cardiac cells or *btl* in MZ progenitors leads to a decrease in both DomeMESO-RFP and Col/*tep4* expression, establishing that both progenitor types are affected. We modified the text in order to clarify this point.

Reviewer #3:In this manuscript, Destalminil-Letourneau et al. describe a novel mechanism of progenitor maintenance in the hematopoietic organ, the lymph gland. They provide evidence for the presence of a vascular niche via cardiac cells, that regulates blood progenitor maintenance in the lymph gland. This manuscript provides interesting and novel insights by showing that the existence of a vascular niche as a conserved mechanism of blood stem cell maintenance as it also exists in flies. The authors provide mechanistic insight by providing data that supports the assertion that vascular niche-lymph gland mediated FGF pathway (via Bnl-Btl) signalling positively regulates hematopoietic progenitor maintenance by controlling intracellular calcium levels in the medullary zone of the lymph gland. The manuscript reports some exciting findings but could benefit from a few improvements requiring further experimentation, validation, and analysis. Major comments are as follows:1) In Figure 2A-A' the authors that the Bnl signal is present throughout the primary lymph gland lobe. This raises the possibility that cells in Posterior Signaling Centre or the Cortical Zone can also be a source of Bnl for the prohemocytes via a type of reciprocal signalling. One way to look at the possibility that the PSC cells or CZ cells are a potential source of Bnl is to include high resolution images where Bnl is detected (either via antibody or in situ) and simultaneously label PSC cells (with collier or Antp-Gal4 driven mCD8GFP) or CZ cells (for example with Hmldelta-Gal4 driven mCD8GFP) with an in-situ against Bnl or with Bnl antibody – to rule out the possibility.

A detailed expression profile of Bnl is provided. Since Bnl is diffusible, we performed ISH with simultaneous immunostainings with specific markers for the different lymph gland cell types (PSC, MZ, crystal cell and differentiating hemocytes (Hml>GFP)). These new data are provided in Figure 2-A-A” and in Figure 2—figure supplement 1A-C”. We also looked at endogenous Bnl expression using the bnl:GFP^endo^ knock-in alleles described by Du et al., 2018 (please see Figure 2S-T’). For Btl, we used the btl:mcherry^endo^ knock-in allele generated by Du et al., 2018 and performed co-staining with markers for different lymph gland cell types (PSC, MZ, crystal cell and differentiating hemocytes (Hml>GFP)). These new data are provided in Figure 3A-A” and Figure 3—figure supplement 1 A-C”.

2) The author's arguments about tissue specific requirement of Bnl would be strengthened by testing whether the increased differentiation in bnlP2 mutants can be rescued by restoring bnl levels by transgene-mediated expression in the cardiac tube and the MZ. For example it could be determined whether the constitutive activation of Btl in prohemocytes in the genetic background of Btl mutants used (btldev1/+) rescues the crystal cell and plasmatocyte differentiation. Additionally, the authors could check if expressing Bnl in the MZ using dome or tep4-Gal4 can rescue the BnlRNAi phenotype of increased prohemocyte differentiation.

We expressed simultaneously *bnl-RNAi* and *bnl* in MZ progenitors (*dome-gal4>bnl-RNAi>bnl*) and observed a rescue in crystal cell numbers compared to the reduction of *bnl* alone (please see the quantification below). This result confirms that *bnl-RNAi* is specific to *bnl*, but this result does not say much about the relative contribution of *bnl* provided by MZ progenitors and *bnl* secreted by cardiac cells since we perform an overexpression of *bnl* in the MZ. As we already include many supplementary figures and as this information is not essential we did not integrate this result in the manuscript. However, what is more informative relative to the *bnl* source (CT versus MZ) are the novel data given in Figure 2S-U. We looked at endogenous Bnl using the bnl:GFP^endo^ knock-in allele (Du et al., 2018). When *bnl* was knocked-down in the cardiac tube (*handD>bnl RNAi, bnl:GFP*^endo^ ), lower Bnl:GFP^endo^ levels were recorded in MZ progenitors (Figure 2S-U). This result illustrates that Bnl secreted by cardiac cells contributes to Bnl levels in progenitors. These novel data are provided in Figure 2 and discussed in the text.

3) Figure 3A also shows strong expression of Btl receptor in the cells towards the cortical zone (periphery of the LG). This raises some intriguing mechanistic questions. First, the manuscript would greatly benefit with a better, systematic, analysis of Bnl and Btl expression both in each of the zonal compartments with appropriate markers (PSC, MZ and CZ) – with high magnification/high resolution images. Second, One can envision a scenario where are the cells in the CZ themselves act as a source of Bnl that binds to Btl in neighboring (MZ) cells to regulate differentiation. Currently the authors have not ruled out this alternative model. If the Bnl is indeed secreted by differentiated cells then there could be 2 modes of signalling – either cell autonomous/paracrine mode or reciprocal signalling to maintain the MZ. It would be helpful if further clarification is provided of the role played by Bnl/Btl-FGF signalling in the CZ. It is possible is that the Bnl secreted by the cardiac cells is transcytosed/transported to the differentiated cells where it regulates differentiation cell autonomously. The authors could test this by perturbing Bnl and Btl in the CZ (using Hmldelta-Gal4 or eater-Gal4 for plasmatocytes and lz-Gal4 for crystal cells) and asking whether this affects crystal cell or plasmatocyte differentiation cell autonomously.

We have now established a detailed expression pattern of *bnl* and Btl using markers for the different LG compartments (please see above our response to point 1). Following the reviewer’s recommendation, we have studied the consequence of reducing bnl levels in crystal cells and in differentiating hemocytes with the specific Lz-Gal4 and HmlGal4 drivers, respectively. A decrease in mature crystal cells (labelled by Hnt) and mature plasmatocytes (labelled by P1) is observed when Lz-gal4 and Hml-gal4, respectively, are used to express bnl-RNAi (please see below the data). This result reveals a cell-autonomous function of *bnl* in these cells. These phenotypes are opposite to those due to *bnl* or *btl* reduction in cardiac cells and MZ progenitors, respectively. Multiple and distinct functions depend on the cellular context. Since this manuscript is centered on the communication between the cardiac tube and MZ progenitors, for the sake of clarity and to avoid a dilution of the main message, we choose to not present these data in the revised manuscript.

4) As noted by the authors it was previously shown that another FGF pathway in *Drosophila*, mediated by Heartless, is required for regulating hematopoiesis in the lymph gland (Dragojlovic-Munther et al., 2013). In this previous report overexpressing Btl in the progenitors had no effect (Figure S6B and K of that paper), this merits some discussion ad explanation from the authors.

In the paper Dragojlovic-Munther et al., 2013, no data were reported relative to the overexpression of the receptor breathless (Btl) in lymph gland progenitors. For Figure S5B of this paper, the authors performed the overexpression of the ligand *bnl* in progenitors. They measured the % of *pxn* labelled cells (*pxn* is a Cortical Zone marker) relative to all lymph gland cells and did not observe a significant difference compared to control. Unfortunately they did not analyze crystal cell or MZ progenitor indexes which might have been more sensitive reporters of the phenotype than *pxn* expression.

In our study, we used 2 independent UAS-Bnl transgenic lines (UASbnl described in Jarecki et al., 1999 and UAS-Bnl::GFP from Lin and Affolter, 2009) and the UAS-Btl^CA^ (Pares and Ricardo, 2016). All 3 gave similar results (see Figure 2I; Figure 3G and Figure 2—figure supplement 2I).

Furthermore, since both Btl and Heartless signalling work through the transcription effector gene pointed we would expect cross talk between the two FGF pathways. This raises an important question of how the two pathways coordinated in the lymph gland. Some genetic interaction analysis between the components of the two arms of FGF signalling will be particularly useful. Specifically, determining if one of the FGF pathways is epistatic to the other, as well as establishing whether indeed both pathways converge on Pointed.

Thank you to the reviewer for her/his suggestion. To investigate the hierarchy between Htl/FGF and Btl/FGF pathways in MZ progenitors we performed epistasis experiments. Our results suggest that there is no hierarchy between these pathways and that both are required simultaneously in MZ progenitors to control lymph gland hematopoiesis. These new data are given in (Figure 3N-P) and are discussed in the text.

5) The Rab11 data in Figure 4B-B' is not as strong as it could be. First, the addition of a cell membrane marker and images of a better and higher resolution would allow greater clarity. Also, since Rab11 marks the recycling endosome, it would be more appropriate to look at other endocytic markers like Rab5 (early endosome marker) and/or Rab7 (late endosome) in addition to Rab11. Also, the co-expression data in the above cases with the Rabs is currently qualitative. It is appropriate in this sort of instance to include quantitative measurement of co-localization. The authors should consider including a staining showing endocytic localization (marked with the Rabs) of the Btl receptor (by in-situ or by antibody) in the prohemocytes. Also in Figure 4B-B' shows only two cells which have Rab11-bnl co-expression/co-localization whereas the field of interest seems to have many more cells in the area. Why don't all the prohemocytes show this co-expression? Finally, also in Figure 4 Bnl diffusion data in a-a' does not look convincing without a cellular marker to identify the hemocyte populations in the LG. Such a marker should be added.

In agreement with the reviewer’s comment we better defined bnl::GFP propagation from cardiac cells to MZ progenitors. We used the ubiRab11-cherryFP reporter and performed Rab7 immunostaining, which label recycling vesicles and late endosomes, respectively. Unfortunately we cannot look at Rab5 vesicles since the Rab5 reporter is tagged by GFP and therefore colocalisation with Bnl::GFP cannot be done. We also performed Col immunostaining to visualize core progenitors. These additional data are provided in Figure 4 A-A”. In addition, we looked at Bnl::GFP co-localisation with the Btl receptor (Figure 4 B-B”) and finally at Bnl::GFP diffusion from the CT to the CZ when the secretion of cardiac cells was impaired by *sar1-RNAi* expression. These data are provided in Figure 4 E-G. Altogether, they strongly support the proposition that Bnl::GFP secreted by cardiac cells is internalized in MZ progenitors likely through receptor-mediated endocytosis. This is discussed in the revised version of the manuscript.